# AlignPrune: Robust Dynamic Data Pruning through Loss Trajectory Alignment

## Abstract

Existing dynamic data pruning methods often fail under noisy-label settings, as they typically rely on per-sample loss as the ranking criterion. This could mistakenly lead to preserving noisy samples due to their high loss values, resulting in significant performance drop. To address this, we propose **AlignPrune**, a noise-robust module designed to enhance the reliability of dynamic pruning under label noise. Specifically, AlignPrune introduces the Dynamic Alignment Score (DAS), which is a loss-trajectory-based criterion that enables more accurate identification of noisy samples, thereby improving pruning effectiveness. As a simple yet effective plug-and-play module, AlignPrune can be seamlessly integrated into state-of-the-art dynamic pruning frameworks, consistently outperforming them without modifying either the model architecture or the training pipeline. Extensive experiments on five widely-used benchmarks across various noise types and pruning ratios demonstrate the effectiveness of AlignPrune, boosting accuracy by up to 6.3% over state-of-the-art baselines. Our results offer a generalizable solution for pruning under noisy data, encouraging further exploration of learning in real-world scenarios.

## 1 Introduction

Modern deep learning models are increasingly trained on large-scale datasets, resulting in substantial computational demands (Radford et al., 2021; Oquab et al., 2023; Fan et al., 2025; Achiam et al., 2023; Brown et al., 2020). To mitigate this, *data pruning* (Guo et al., 2022) has emerged as an effective strategy for reducing total training cost by discarding low-utility samples, while without degrading the final model performance.

Existing data pruning approaches fall into two main categories, the first being *static coreset selection* (Toneva et al., 2018; Paul et al., 2021; Park et al., 2022; Han et al., 2018; Coleman et al., 2019; Sener & Savarese, 2018; Sorscher et al., 2022; Xia et al., 2022; Zhang et al., 2024), which pre-selects a fixed subset of the full dataset before training begins.

However, such static approaches are often brittle in real-world scenarios. The challenge of label noise has become increasingly crucial in the current era of large-scale models, where the immense cost of curating perfectly clean data presents a major bottleneck. Modern large-scale datasets are therefore often noisy (Kisel et al., 2024), due to weak image-text supervision (Radford et al., 2021), automatic web crawlers (Oquab et al., 2023), or inherent semantic ambiguity (Singh et al., 2024). Once a noisy or uninformative sample is included in the selected subset, it remains in the training loop permanently, inevitably degrading the model's performance.

*Dynamic data pruning* (Raju et al., 2021; Qin et al., 2024; Zhou et al., 2025) offers a more robust alternative by adaptively updating the subset selection during model training. By exposing the model to a broader portion of the dataset throughout training, this paradigm allows for the revision of which samples to keep or discard, making it inherently more suitable under noisy scenarios.

Despite this promise, current state-of-the-art dynamic pruning methods, such as InfoBatch (Qin et al., 2024) and SeTa (Zhou et al., 2025), possess a critical vulnerability: they typically rank each training sample based on their individual loss values. This criterion becomes unreliable under label noise, as noisy samples often tend to produce large losses (Jiang et al., 2018). Consequently, existing methods mistakenly preserve or even prioritize noisy samples during the pruning process, resulting in degraded performance under noisy-label scenarios.

To address this limitation, we are the first to investigate dynamic data pruning under label noise, proposing *AlignPrune*, a noise-robust and effective plug-and-play module for improving the reliability of dynamic pruning. The key idea is to replace the loss-based ranking criteria with a noisy-robust signal: the proposed Dynamic Alignment Score (DAS), which quantifies the correlation between each sample's loss trajectory and the average loss from a reference set over time.

The core intuition is that clean and noisy samples exhibit fundamentally different learning dynamics. Clean samples tend to follow a consistent pattern of loss decay as the model improves, while noisy samples exhibit more erratic or inconsistent behavior. By comparing these temporal patterns, DAS can effectively provide a robust signal for accurately identifying and filtering out noisy data samples.

AlignPrune can be seamlessly integrated into existing dynamic pruning methods (Qin et al., 2024; Zhou et al., 2025), by replacing their loss-based sample ranking with our loss-trajectory-based criterion. The resulting framework preserves the unbiasedness of the gradient expectation of vanilla dynamic pruning method, while improving resilience to noise without requiring any change to the model architecture or the training pipeline.

We evaluate AlignPrune on five widely-used benchmarks: CIFAR-100N, CIFAR-10N (Wei et al., 2021), WebVision (Li et al., 2017), Clothing-1M (Xiao et al., 2015) and ImageNet-1K (Deng et al., 2009), under a wide range of noisy-label conditions. Experiments demonstrate that AlignPrune consistently outperforms both static and dynamic baselines in noisy-label settings, while maintaining competitive performance under the clean-label case. Our method also improves training efficiency, yielding higher accuracy while also reducing the total time cost.

## 2 RELATED WORK

**Static Data Pruning.** Static pruning methods (Guo et al., 2022; Park et al., 2022; Han et al., 2018; Sorscher et al., 2022; Xia et al., 2022; Zhang et al., 2024) aim to select a fixed subset of the original dataset that yields comparable performance to training on the whole dataset. Existing approaches rely on pre-defined or heuristic metrics, which can be broadly categorized into several types: informative-based (Tan et al., 2025; 2023), error-based (Toneva et al., 2018; Paul et al., 2021), geometry-based (Agarwal et al., 2020; Sener & Savarese, 2018), gradient-matching-based (Mirzasoleiman et al., 2020a; Killamsetty et al., 2021a), uncertainty-based (Coleman et al., 2019; He et al., 2024; Cho et al., 2025), coverage-based (Zheng et al., 2022; 2025), bilevel-optimization-based (Killamsetty et al., 2021b), memorization-based (Agiollo et al., 2024; Garg & Roy, 2023), sub-modularity-based methods (Iyer et al., 2021). More recently, Nagaraj et al. (2025) introduce the concept of using loss trajectories for static coreset selection, which is conceptually related to our approach. However, their method is limited to clean-label settings and only static pruning scenarios, which also lacks a theoretically grounded justification despite achieving lossless empirical results.

**Dynamic Data Pruning.** Dynamic pruning methods aim to reduce total training cost by selecting a sequence of training subsets over epochs, while maintaining model performance. These methods typically perform epoch-wise pruning based on a scoring criterion. For example, Raju et al. (2021) leverages a dynamic uncertainty estimation, Qin et al. (2024) and Zhou et al. (2025) rely directly on per-sample loss values for ranking. IES (Yuan et al., 2025) utilizes second-order loss dynamics to identify informative samples, and Yang et al. (2025) unifies dynamic data selection and augmentation under the dynamic pruning. Some methods (Wu et al., 2025; Hong et al., 2024) apply sample selection within each training batch. Recently, Guo & Kankanhalli (2024) and Zhao et al. (2025) extend this dynamic pruning strategy to contrastive pre-training frameworks. Despite promising results, none of the above explore dynamic pruning under noisy-label settings. Current approaches remain vulnerable in such scenarios due to their reliance on non-robust ranking signals, which often lead to the repeated selection and training of noisy samples.

**Noisy Label Learning.** Noisy Label Learning (Song et al., 2022; Li et al., 2022; Zhang et al., 2023; Lyu & Tsang, 2019; Li et al., 2020; Liu et al., 2020; 2022a; Park et al., 2023) has been extensively studied to improve the robustness of deep networks in real-world settings. Recent methods primarily adopt self-supervised or semi-supervised techniques to re-label noisy data via consistency regularization (Li et al., 2020; Liu et al., 2020; 2022a; Park et al., 2023). For example, DivideMix (Li et al., 2020) employs co-training with a dynamic mixture model, ELR+ (Liu et al.,

2020) introduces entropy-based regularization, and SOP+ (Liu et al., 2022a) incorporates label smoothing with learnable consistency targets. In addition, Prune4ReL (Park et al., 2023) proposes a noise-aware pruning method, aiming to select subsets to maximize re-labeling accuracy and downstream generalization performance. While effective, these methods often require substantial computation overhead or multiple-model training, limiting their practicality in large-scale settings. However, data pruning methods offers a natural solution to scalability by reducing the number of samples processed while maintaining or even improving training performance. Our approach builds on this idea by integrating data pruning techniques into noisy label learning.

# 3 METHODOLOGY

In this section, we present *AlignPrune*, a plug-and-play dynamic data pruning module designed to improve model robustness under label noise.

## 3.1 PRELIMINARIES

We focus on the task of data pruning for robust classification task under noisy label scenarios, which is a setting widely studied in the machine learning community. Given a noisy dataset $\mathcal{D} = \{z_i\}|_{i=1}^{|\mathcal{D}|} = \{(x_i, \tilde{y}_i)\}|_{i=1}^{|\mathcal{D}|}$, where $x_i \in \mathcal{X} \subset \mathbb{R}^d$ denotes the input and $\tilde{y}_i$ denotes the corresponding label, which is possibly noisy. The goal of data pruning is to find a smaller but most informative subset $\mathcal{S} \subset \mathcal{D}$ with $|\mathcal{S}| < |\mathcal{D}|$ [1], such that a model $\theta_{\mathcal{S}}$ trained on $\mathcal{S}$ achieves comparable or better performance than the model $\theta_{\mathcal{D}}$ trained on the full dataset $\mathcal{D}$. Formally, we aim to find the optimal subset $\mathcal{S}^*$ satisfying:

$$\mathcal{S}^* = \underset{\mathcal{S}: \, |\mathcal{S}| \leq s}{\mathrm{argmax}} \sum_{(x,\tilde{y}) \in \mathcal{D}} \mathbb{1}_{[f(x;\theta_{\mathcal{S}}) = y^*]} \; : \; \theta_{\mathcal{S}} = \underset{\theta}{\mathrm{argmin}} \; \mathcal{L}(\theta; \mathcal{S}), \tag{1}$$

where $f(x; \theta_{\mathcal{S}}) \in \mathbb{R}^c$ is the model's prediction vector, $y^*$ is the ground-truth label of a noisy example $x$, and $s$ is the target subset size.

However, finding the optimal subset $\mathcal{S}^*$ through direct optimization of Eq. 1 is infeasible in practice, because the ground-truth labels $y^*$ are unknown. Therefore, existing approaches (Park et al., 2023; Qin et al., 2024) often rely on heuristics (loss values, or certain conditions), to identify informative or clean examples. However, these methods break down in the presence of noisy labels. To address this, we propose a robust, dynamic sample-ranking criterion in the next section, which estimates label correctness based on trajectory-level consistency with clean behavior. This enables reliable pruning without needing the correct ground-truth label $y^*$.

## 3.2 DYNAMIC ALIGNMENT SCORE

Previous dynamic data pruning methods (Raju et al., 2021; Qin et al., 2024; Zhou et al., 2025) typically rely on per-epoch training loss to rank the importance of each sample. While this is effective in clean-label settings with grounded theoretical supports in Qin et al. (2024), it becomes unreliable with the introduce of label noise. Noisy samples often incur large losses and are mistakenly preserved during training (Jiang et al., 2018), leading to degraded performance. To address this issue, we propose a new ranking criterion named the Dynamic Alignment Score (DAS), which evaluates the correctness of the label of training sample by measuring the consistency of its loss trajectory with a clean reference signal.

**Loss Trajectory.** To capture the training dynamics of each sample, we define a sample's loss trajectory as the sequence of its training loss values over a fixed window of recent epochs. Let $\ell_i^{(t)}$ denote the training loss of $i$-th sample at epoch $t$, computed as:

$$\ell_i^{(t)} = \mathcal{L}(f_{\theta_t}(x_i), \tilde{y}_i), \tag{2}$$

---

[1]For static pruning, we aim to identify the most representative subset $\mathcal{S}$ from $\mathcal{D}$ according to specific selection criterion before training. While for dynamic pruning, we aim to identify a sequence of subsets $\mathcal{S} = \{\mathcal{S}_t\}_{t=1}^T$ across each training epoch, where $\mathcal{S}_t \subset \mathcal{D}$ with $|\mathcal{S}_t| < |\mathcal{D}|$ for each epoch $t$.

where $\theta_t$ is the trained model at epoch $t$, $f_{\theta_t}(x_i)$ is model's prediction, and $\tilde{y}_i$ is the (potentially noisy) label. Given a trajectory window size of $N$ epochs, the loss trajectory vector $\mathbf{v}_i^{(t)} \in \mathbb{R}^T$ for the $i$-th sample at epoch $t$ is defined as:

$$\mathbf{v}_i^{(t)} = \left[\ell_i^{(t-N+1)}, \ell_i^{(t-N+2)}, \ldots, \ell_i^{(t)}\right]. \tag{3}$$

To obtain a clean reference, we utilize a small held-out validation set $\mathcal{D}_{ref}$, which is assumed to be free of label noise [2]. For each epoch $t'$, we compute the average reference loss and the resulting reference loss trajectory $\mathbf{v}_{ref}^{(t')} \in \mathbb{R}^T$ as:

$$\bar{\ell}_{ref}^{(t')} = \frac{1}{|\mathcal{D}_{ref}|} \sum_{(x_j, \tilde{y}_j) \in \mathcal{D}_{ref}} \mathcal{L}(f_{\theta_{t'}}(x_j), \tilde{y}_j) \ , \ \ \mathbf{v}_{ref}^{(t')} = \left[\bar{\ell}_{ref}^{(t'-N+1)}, \bar{\ell}_{ref}^{(t'-N+2)}, \ldots, \bar{\ell}_{ref}^{(t')}\right]. \tag{4}$$

**Correlation-Based Alignment.** With the loss trajectories of training samples $\mathbf{v}_i$ and the clean validation reference $\mathbf{v}_{ref}$ defined, we now introduce the *Dynamic Alignment Score* (DAS), which provides a correlation-based metric that quantifies how closely each training sample aligns with the overall trend of clean reference performance. The key intuition is that samples with correct labels $\tilde{y} = y^*$ typically follow a consistent trajectory of decreasing loss that mirrors the behavior of a clean reference set, while noisy samples exhibit more erratic or inconsistent patterns. The proposed DAS quantifies this alignment by computing the correlation score between the training sample's loss trajectory and the reference trajectory.

Formally, given a training sample's loss trajectory $\mathbf{v}_i^{(t)}$ and the clean validation reference $\mathbf{v}_{ref}^{(t)}$ at epoch $t$, we define the Dynamic Alignment Score as:

$$DAS_i^{(t)} = \rho(\mathbf{v}_i^{(t)}, \mathbf{v}_{ref}^{(t)}), \tag{5}$$

where $\rho$ denotes the correlation function. In our implementation, Pearson's correlation (Pearson, 1895) is used due to its scale invariance and computational efficiency.

A positive DAS value indicates synchronized learning dynamics with the expected clean loss pattern, suggesting the training sample is likely to be clean. A negative DAS value, on the other hand, reflects a conflicting learning dynamics with clean reference behavior, implying the presence of label noise.

In practice, DAS is computed in batch using vectorized matrix operations and updated at the end of each epoch. The per-sample trajectories are stored in a memory bank of window size $N$, allowing for efficient computation without excessive overhead. While we use Pearson correlation as the default $\rho$, other similarity metrics will be discussed further in the ablation studies in Sec. 4.

### 3.3 ALIGNPRUNE: A NOISE-ROBUST DYNAMIC PRUNING MODULE

Building on the *Dynamic Alignment Score* (DAS), we now describe how to integrate it into existing dynamic data pruning frameworks. Our proposed module, *AlignPrune*, serves as a plug-and-play replacement for loss-based ranking in dynamic methods such as InfoBatch (Qin et al., 2024) and SeTa (Zhou et al., 2025), enabling improved robustness under noisy label scenarios.

In prior dynamic pruning methods designed for clean labels, each training sample is ranked based on its per-epoch loss. The pruning is then performed either by discarding samples below the mean loss (Qin et al., 2024), or via a sliding-window strategy to remove ineffective sample over time (Zhou et al., 2025). However, as discussed in Sec. 3.2, loss values alone are unreliable indicators under label noise. To address this, *AlignPrune* replaces the loss-based ranking metric with the proposed DAS as:

$$score_i^{(t)} := DAS_i^{(t)}, \tag{6}$$

where a higher score indicates stronger alignment with clean reference behavior, and thus a higher likelihood of the sample being clean. Training on subsets with higher DAS values enables the model to focus on more reliable samples and mitigates the negative impact of noisy labels. The complete algorithm of the plug-and-play *AlignPrune* is outlined in Algorithm 1 below.

---

[2]We treat this as an idealized assumption and will empirically evaluate both the size and purity of this reference set in Sec. 4.4.2.

---

**Algorithm 1** AlignPrune for Dynamic Data Pruning

---

INPUT: Noisy training set $\mathcal{D}$, Clean reference set $\mathcal{D}_{ref}$, Dynamic data pruning method $\mathcal{M}$ (with pruning probability $r$ [a]), Total epochs $T$, Trajectory window size $N$, Correlation function $\rho$

1: Initialize model parameters $\theta^{(0)}$
2: **for** epoch $t = 1, ..., T$ **do**
3:    **for** each sample $z_i \in \mathcal{S}^{(t)} \subset \mathcal{D}$ **do**
4:       Compute training loss $\ell_i^{(t)}$ by Eq. 2
5:       Update trajectory $\mathbf{v}_i^{(t)}$ with $\ell_i^{(t)}$ by Eq. 3
6:    **end for**
7:    Compute average reference loss $\bar{\ell}_{ref}^{(t)}$ over $\mathcal{D}_{ref}$ by Eq. 4
8:    Update reference trajectory $\mathbf{v}_{ref}^{(t)}$ with $\bar{\ell}_{ref}^{(t)}$ by Eq. 4
9:    Compute $DAS_i^{(t)} = \rho(\mathbf{v}_i^{(t)}, \mathbf{v}_{ref}^{(t)})$ for each $z_i \in \mathcal{S}^{(t)}$
10:   Update $\theta^{(t)}$ on $\mathcal{S}^{(t)}$ following the update rule of $\mathcal{M}$ [b]
11:   Apply pruning: $\mathcal{S}^{(t+1)} \leftarrow \mathcal{M}(\mathcal{S}^{(t)}, DAS, r)$
12: **end for**
OUTPUT: Final trained model $\theta^{(T)}$

---

[a]Pruning probability $r$ is a hyper-parameter used in dynamic pruning methods; see details in Sec. 4.1.

[b]The specific gradient update rule is determined by the base pruning method $\mathcal{M}$. For instance, InfoBatch uses an expectation rescaling operation to ensure unbiased gradients, which is preserved when using AlignPrune.

Importantly, *AlignPrune* modifies only the sample ranking strategy while preserving the base model architecture, original training loop, and previous pruning mechanism, ensuring seamless integration to existing dynamic pruning frameworks. In Sec. 4, we demonstrate that this simple modification consistently improves performance across multiple datasets and various noisy label conditions, while maintaining training efficiency.

## 4 EXPERIMENTS

### 4.1 EXPERIMENTAL SETUP

**Datasets.** We evaluate *AlignPrune* on four benchmark datasets commonly used in noisy-label research: **CIFAR-100N**, **CIFAR-10N** (Wei et al., 2021), **WebVision** (Li et al., 2017) and **Clothing-1M** (Xiao et al., 2015). CIFAR-N datasets facilitate controlled studies under both *synthetic* and *real* noise conditions, while WebVision and Clothing-1M offer large-scale settings with real-world noise. We include **ImageNet-1K** (Deng et al., 2009) with *synthetic* noise for further large-scale experiments.

**Baseline Methods.** We compare *AlignPrune* with static and dynamic pruning methods, along with their respective random selection baselines. For static pruning, we consider representative methods including: SmallL (Jiang et al., 2018), Margin (Coleman et al., 2019), Forgetting (Toneva et al., 2018), GraNd Paul et al. (2021), Moderate (Xia et al., 2022), SSP (Sorscher et al., 2022), Prune4ReL (Park et al., 2023), Dyn-Unc (He et al., 2024) and DUAL Cho et al. (2025). For dynamic pruning, we benchmark two state-of-the-art methods: InfoBatch (Qin et al., 2024) and SeTa (Zhou et al., 2025). Additional discussion on the baseline selection is provided in Appendix A.

**Implementation Details.** For each of the data pruning baselines, we adopt the default optimal hyper-parameter settings to ensure fairness and reproducibility. As for our proposed *AlignPrune*, we align the pruning probability $r$ [3] with the value used in the underlying dynamic pruning methods (Qin et al., 2024; Zhou et al., 2025). $N$ is set to 25 by default, and reduced to 5 for Clothing-1M due to its shorter fine-tuning duration. Pearson's correlation is used as the default $\rho$. For dataset with an official validation split, we use this official split as the clean reference set. For dataset with only train/test splits, we randomly sample a subset from the training split (with equal size to the test split) as the clean reference set. All experiments are conducted on NVIDIA RTX A6000 GPUs. Additional training and reproducibility details are provided in Appendix A.

---

[3]To avoid confusion, we denote the pruning probability $r$ as the fraction of low-score samples discarded in each epoch, and the pruning ratio / prune ratio as the target percentage of the entire dataset that is pruned over the training process, both following the terminology in Qin et al. (2024).

Table 1: **Classification results on CIFAR-100N with ResNet-18.** Performance gaps relative to the full-training setting are indicated as superscripts. The mean $\Delta$ is computed across all noise types.

| Noisy Type → 
 Pruning Method ↓ | Clean | Real | Symmetric | | | Asymmetric | | Mean |
|---|---|---|---|---|---|---|---|---|
| | | | 0.2 | 0.5 | 0.8 | 0.2 | 0.4 | $\Delta$ |
| Full-training | 78.2 | 56.1 | 71.4 | 58.6 | 39.8 | 72.4 | 63.3 | – |
| **Prune Ratio ∼30%** | | | | | | | | |
| Static Random | 73.8 $^{-4.4}$ | 53.3 $^{-2.8}$ | 67.9 $^{-3.4}$ | 51.1 $^{-7.5}$ | 30.3 $^{-9.5}$ | 68.4 $^{-4.0}$ | 57.8 $^{-5.5}$ | -5.3 |
| Dynamic Random | 77.3 $^{-0.9}$ | 54.7 $^{-1.4}$ | 69.9 $^{-1.4}$ | 58.8 $^{+0.2}$ | 40.1 $^{+0.3}$ | 71.5 $^{-0.9}$ | 63.2 $^{-0.1}$ | -0.6 |
| InfoBatch (Qin et al., 2024) | 79.0 $^{+0.8}$ | 56.1 $^{+0.0}$ | 71.4 $^{+0.0}$ | 59.7 $^{+1.1}$ | 41.8 $^{+2.0}$ | 71.9 $^{-0.5}$ | 64.2 $^{+0.9}$ | +0.6 |
| **InfoBatch + Ours** | 79.3 $^{+1.1}$ | 59.4 $^{+3.3}$ | 71.8 $^{+0.4}$ | 66.0 $^{+7.4}$ | 41.8 $^{+2.0}$ | 72.6 $^{+0.2}$ | 68.0 $^{+4.7}$ | **+2.7** |
| SeTa (Zhou et al., 2025) | 79.0 $^{+0.8}$ | 55.6 $^{-0.5}$ | 70.2 $^{-1.2}$ | 59.0 $^{+0.4}$ | 41.6 $^{+1.8}$ | 71.4 $^{-0.9}$ | 63.2 $^{-0.1}$ | +0.0 |
| **SeTa + Ours** | 79.3 $^{+1.1}$ | 56.3 $^{+0.2}$ | 70.8 $^{-0.5}$ | 60.5 $^{+1.9}$ | 41.6 $^{+1.8}$ | 71.9 $^{-0.5}$ | 64.3 $^{+1.0}$ | **+0.7** |
| **Prune Ratio ∼50%** | | | | | | | | |
| Static Random | 72.1 $^{-6.1}$ | 51.8 $^{-4.3}$ | 64.0 $^{-7.4}$ | 47.2 $^{-11.5}$ | 22.9 $^{-16.9}$ | 65.3 $^{-7.0}$ | 54.6 $^{-8.7}$ | -8.8 |
| Dynamic Random | 75.3 $^{-2.9}$ | 54.1 $^{-2.0}$ | 70.4 $^{-1.0}$ | 59.5 $^{+0.9}$ | 40.7 $^{+0.9}$ | 70.1 $^{-2.3}$ | 64.8 $^{+1.6}$ | -0.7 |
| InfoBatch (Qin et al., 2024) | 77.7 $^{-0.5}$ | 56.0 $^{-0.1}$ | 71.3 $^{-0.1}$ | 60.5 $^{+1.9}$ | 42.2 $^{+2.4}$ | 71.8 $^{-0.6}$ | 65.2 $^{+2.0}$ | +0.7 |
| **InfoBatch + Ours** | 78.5 $^{+0.3}$ | 60.7 $^{+4.6}$ | 71.6 $^{+0.3}$ | 62.0 $^{+3.4}$ | 42.6 $^{+2.8}$ | 72.6 $^{+0.3}$ | 68.6 $^{+5.3}$ | **+2.4** |
| SeTa (Zhou et al., 2025) | 77.5 $^{-0.7}$ | 55.7 $^{-0.4}$ | 70.7 $^{-0.7}$ | 60.0 $^{+1.4}$ | 40.5 $^{+0.7}$ | 71.9 $^{-0.5}$ | 64.5 $^{+1.2}$ | +0.2 |
| **SeTa + Ours** | 78.4 $^{+0.2}$ | 56.3 $^{+0.1}$ | 71.2 $^{-0.2}$ | 61.0 $^{+2.4}$ | 41.9 $^{+2.1}$ | 72.2 $^{-0.2}$ | 66.0 $^{+2.7}$ | **+1.0** |
| **Prune Ratio ∼70%** | | | | | | | | |
| Static Random | 69.7 $^{-8.5}$ | 45.9 $^{-10.2}$ | 56.7 $^{-14.6}$ | 39.0 $^{-19.6}$ | 15.8 $^{-24.0}$ | 58.4 $^{-13.9}$ | 49.5 $^{-13.8}$ | -14.9 |
| Dynamic Random | 75.2 $^{-3.0}$ | 53.3 $^{-2.9}$ | 70.2 $^{-1.2}$ | 62.7 $^{+4.1}$ | 41.0 $^{+1.2}$ | 71.9 $^{-0.5}$ | 67.3 $^{+4.0}$ | +0.3 |
| InfoBatch (Qin et al., 2024) | 77.1 $^{-1.1}$ | 55.0 $^{-1.2}$ | 71.4 $^{+0.1}$ | 63.6 $^{+5.0}$ | 42.4 $^{+2.6}$ | 72.2 $^{-0.2}$ | 67.8 $^{+4.5}$ | +1.4 |
| **InfoBatch + Ours** | 77.5 $^{-0.7}$ | 58.3 $^{+2.2}$ | 72.2 $^{+0.9}$ | 64.7 $^{+6.1}$ | 42.8 $^{+3.0}$ | 72.5 $^{+0.1}$ | 69.1 $^{+5.9}$ | **+2.5** |
| SeTa (Zhou et al., 2025) | 77.2 $^{-1.1}$ | 55.2 $^{-1.0}$ | 71.6 $^{+0.2}$ | 62.4 $^{+3.8}$ | 42.4 $^{+2.6}$ | 72.0 $^{-0.4}$ | 67.4 $^{+4.1}$ | +1.2 |
| **SeTa + Ours** | 77.8 $^{-0.4}$ | 55.5 $^{-0.6}$ | 72.3 $^{+0.9}$ | 63.3 $^{+4.7}$ | 42.7 $^{+2.9}$ | 72.7 $^{+0.3}$ | 67.6 $^{+4.3}$ | **+1.7** |

**Evaluation.** Following prior work (Qin et al., 2024; Zhou et al., 2025), we evaluate each method at pruning ratios of {30%, 50%, 70%} for all the datasets. For dynamic pruning methods, the pruning ratio is controlled by adjusting the number of training epochs to match the total number of forward passes in the full-training baseline, as in Qin et al. (2024). Each experiment is *repeated three times*, and we report the average accuracy from the final epoch.

## 4.2 Main Results

**Performance Comparisons.** Table 1 and Table 2 present the test accuracy of baseline methods and AlignPrune on CIFAR-100N and CIFAR-10N under three pruning ratios. [4] Overall, Align-Prune consistently achieves the best performance across all settings, demonstrating strong robustness under various types of label noise. Numerically, on CIFAR-100N, InfoBatch with AlignPrune can achieve an average accuracy improvement

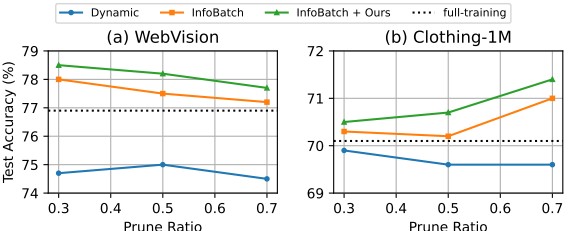

Figure 1: **Data pruning performance comparison: (a) WebVision; (b) Clothing-1M.** Best viewed in color.

of +2.7% under 30% pruning ratio across all noise types, compared to only +0.6% improvement from vanilla InfoBatch. On CIFAR-10N, SeTa with AlignPrune can also yield an average +0.1% improvement under 50% pruning ratio, whereas vanilla SeTa shows a significant drop of -1.4% in the same setting. These results demonstrates that AlignPrune can be effectively integrated into dynamic pruning methods to improve performance under both real and synthetic label noise, without sacrificing the effectiveness when dealing with clean data. See Table I for statistical analysis.

To further validate our approach, Fig. 1 presents a comparison between dynamic pruning baselines and AlignPrune on two large-scale datasets: WebVision and Clothing-1M. Due to instability, the

---

[4]A comprehensive comparison including both static and dynamic pruning methods against AlignPrune is provided in Tables B and C in Appendix B.1.

Table 2: **Classification results on CIFAR-10N with ResNet-18.** Performance gaps relative to the full-training setting are indicated as superscripts. The mean $\Delta$ is computed across all noise types.

| Noisy Type → Pruning Method ↓ | Clean | Real | | Symmetric | | | Asymmetric | | Mean |
|---|---|---|---|---|---|---|---|---|---|
| | | Real-A | Real-W | 0.2 | 0.5 | 0.8 | 0.2 | 0.4 | $\Delta$ |
| Full-training | 95.6 | 90.7 | 78.3 | 91.3 | 85.4 | 65.6 | 90.6 | 85.8 | – |
| **Prune Ratio $\sim$30%** | | | | | | | | | |
| Static Random | 94.6 $^{-1.0}$ | 89.9 $^{-0.7}$ | 75.0 $^{-3.3}$ | 89.0 $^{-2.3}$ | 81.2 $^{-4.2}$ | 57.6 $^{-8.0}$ | 87.9 $^{-2.8}$ | 82.5 $^{-3.3}$ | -3.2 |
| Dynamic Random | 94.8 $^{-0.8}$ | 90.3 $^{-0.3}$ | 77.3 $^{-1.0}$ | 90.8 $^{-0.5}$ | 84.2 $^{-1.2}$ | 64.8 $^{-0.8}$ | 91.5 $^{+0.9}$ | 84.5 $^{-1.3}$ | -0.6 |
| InfoBatch (Qin et al., 2024) | 95.3 $^{-0.3}$ | 91.0 $^{+0.3}$ | 78.1 $^{-0.3}$ | 92.0 $^{+0.7}$ | 85.6 $^{+0.1}$ | 67.5 $^{+1.9}$ | 91.3 $^{+0.7}$ | 87.0 $^{+1.3}$ | +0.5 |
| **InfoBatch + Ours** | 95.3 $^{-0.3}$ | 91.4 $^{+0.7}$ | 82.0 $^{+3.7}$ | 92.7 $^{+1.4}$ | 87.9 $^{+2.5}$ | 67.8 $^{+2.2}$ | 92.9 $^{+2.3}$ | 90.0 $^{+4.2}$ | **+2.1** |
| SeTa (Zhou et al., 2025) | 95.2 $^{-0.4}$ | 90.6 $^{-0.1}$ | 76.3 $^{-2.0}$ | 91.8 $^{+0.5}$ | 84.7 $^{-0.7}$ | 64.2 $^{-1.5}$ | 90.8 $^{+0.1}$ | 88.4 $^{+2.6}$ | -0.2 |
| **SeTa + Ours** | 95.3 $^{-0.3}$ | 90.7 $^{+0.0}$ | 79.1 $^{+0.8}$ | 92.5 $^{+1.2}$ | 85.0 $^{-0.5}$ | 64.6 $^{-1.0}$ | 91.8 $^{+1.2}$ | 89.3 $^{+3.5}$ | **+0.6** |
| **Prune Ratio $\sim$50%** | | | | | | | | | |
| Static Random | 93.3 $^{-2.3}$ | 88.6 $^{-2.1}$ | 72.4 $^{-5.9}$ | 87.4 $^{-3.9}$ | 77.5 $^{-8.0}$ | 51.4 $^{-14.2}$ | 85.8 $^{-4.9}$ | 77.9 $^{-7.9}$ | -6.1 |
| Dynamic Random | 94.5 $^{-1.1}$ | 89.2 $^{-1.5}$ | 76.1 $^{-2.3}$ | 88.9 $^{-2.4}$ | 83.7 $^{-1.8}$ | 63.3 $^{-2.3}$ | 90.6 $^{+0.0}$ | 83.7 $^{-2.1}$ | -1.7 |
| InfoBatch (Qin et al., 2024) | 95.0 $^{-0.6}$ | 90.5 $^{-0.2}$ | 77.7 $^{-0.6}$ | 92.0 $^{+0.7}$ | 87.5 $^{+2.1}$ | 64.3 $^{-1.3}$ | 92.0 $^{+1.3}$ | 87.5 $^{+1.7}$ | +0.4 |
| **InfoBatch + Ours** | 95.0 $^{-0.6}$ | 91.3 $^{+0.6}$ | 82.4 $^{+4.1}$ | 92.8 $^{+1.5}$ | 87.6 $^{+2.1}$ | 64.5 $^{-1.1}$ | 92.1 $^{+1.5}$ | 89.8 $^{+4.0}$ | **+1.5** |
| SeTa (Zhou et al., 2025) | 94.8 $^{-0.8}$ | 89.4 $^{-1.3}$ | 78.5 $^{+0.2}$ | 91.3 $^{+0.0}$ | 86.0 $^{+0.5}$ | 53.0 $^{-12.7}$ | 91.7 $^{+1.1}$ | 87.9 $^{+2.1}$ | -1.4 |
| **SeTa + Ours** | 94.9 $^{-0.7}$ | 89.7 $^{-1.0}$ | 79.3 $^{+1.0}$ | 91.9 $^{+0.6}$ | 87.6 $^{+2.2}$ | 60.2 $^{-5.5}$ | 92.2 $^{+1.6}$ | 88.5 $^{+2.7}$ | **+0.1** |
| **Prune Ratio $\sim$70%** | | | | | | | | | |
| Static Random | 90.2 $^{-5.4}$ | 86.6 $^{-4.1}$ | 69.0 $^{-9.3}$ | 85.3 $^{-6.0}$ | 73.8 $^{-11.6}$ | 41.4 $^{-24.3}$ | 82.3 $^{-8.3}$ | 75.3 $^{-10.5}$ | -9.9 |
| Dynamic Random | 93.0 $^{-2.6}$ | 87.4 $^{-3.2}$ | 75.9 $^{-2.5}$ | 87.4 $^{-3.9}$ | 84.6 $^{-0.8}$ | 60.2 $^{-5.5}$ | 91.4 $^{+0.8}$ | 86.8 $^{+1.0}$ | -2.1 |
| InfoBatch (Qin et al., 2024) | 94.3 $^{-1.3}$ | 89.9 $^{-0.8}$ | 79.4 $^{+1.0}$ | 92.4 $^{+1.1}$ | 87.0 $^{+1.6}$ | 61.0 $^{-4.7}$ | 92.5 $^{+1.8}$ | 88.3 $^{+2.5}$ | +0.2 |
| **InfoBatch + Ours** | 94.3 $^{-1.3}$ | 90.8 $^{+0.1}$ | 81.0 $^{+2.7}$ | 92.8 $^{+1.5}$ | 87.9 $^{+2.5}$ | 61.1 $^{-4.5}$ | 92.9 $^{+2.3}$ | 89.2 $^{+3.4}$ | **+0.8** |
| SeTa (Zhou et al., 2025) | 94.3 $^{-1.3}$ | 89.5 $^{-1.2}$ | 77.7 $^{-0.6}$ | 92.2 $^{+0.9}$ | 87.1 $^{+1.7}$ | 30.4 $^{-35.2}$ | 92.2 $^{+1.5}$ | 88.1 $^{+2.3}$ | -4.0 |
| **SeTa + Ours** | 94.5 $^{-1.1}$ | 90.0 $^{-0.7}$ | 79.4 $^{+1.1}$ | 92.6 $^{+1.3}$ | 87.7 $^{+2.3}$ | 58.0 $^{-7.6}$ | 92.5 $^{+1.9}$ | 88.5 $^{+2.7}$ | **+0.0** |

training of SeTa collapses on these datasets, hence we report results only for InfoBatch and its AlignPrune variant. In line with the observations on CIFAR-N datasets, AlignPrune outperforms existing baselines across different pruning ratios. Quantitatively, InfoBatch with AlignPrune achieves an average improvement of +0.5% over vanilla InfoBatch, even under the fine-tuning duration of Clothing-1M. These findings reinforce the effectiveness of AlignPrune in dynamic pruning, showcasing its scalability and robustness across large-scale datasets in both from-scratch training and fine-tuning settings.

Furthermore, AlignPrune can be seamlessly integrated with existing robust learning techniques to achieve state-of-the-art performance. Full results are detailed in Tables D and E in Appendix B.1.

**Efficiency Comparisons.** In addition to the performance comparison, we compare the training time cost between baseline methods and AlignPrune. Specifically, we use the default setting to train CIFAR-100N with a pruning ratio of 50% under the Real noisy type. As reported in Table 3, under identical computational conditions, our method not only

Table 3: **Efficiency comparison on CIFAR-100N.** Results are reported with ResNet-18 under 50% prune ratio with Real noise for 200 epochs on two RTX-A6000-GPUs. "Ours" refers to AlignPrune with InfoBatch.

| | Static | Dynamic | SeTa | InfoBatch | **Ours** | Full Data |
|---|---|---|---|---|---|---|
| Acc (%) ↑ | 51.8 | 54.1 | 55.7 | 56.0 | **60.7** | 56.1 |
| Time (mins) ↓ | 24.7 | 24.4 | 25.8 | 23.9 | **22.5** | 46.7 |

improves accuracy but also enhances training efficiency. While the DAS calculation introduces a negligible per-epoch overhead (see Table F in Appendix B.2.1 for a detailed analysis), the resulting improvement in sample selection efficiency leads to a significant reduction in the total training time compared to both the baseline pruning methods and full-data training.

## 4.3 LARGE-SCALE EXPERIMENTAL RESULTS

To further validate the scalability and robustness of AlignPrune, we extend our experiments to ImageNet-1K (Deng et al., 2009) classification under both clean and noisy-label settings, across modern CNN-based (Liu et al., 2022b) and ViT-based (Touvron et al., 2021; Liu et al., 2021) architectures

Table 4: **Classification results on ImageNet-1K across modern CNN and ViT architectures.** Performance gaps relative to the full-training setting are indicated as superscripts.

| Model Architecture → | | ConvNeXt | | DeiT | | Swin | | Mean |
|---|---|---|---|---|---|---|---|---|
| Pruning Method ↓ | | Tiny | Base | Small | Base | Tiny | Base | Δ |
| Clean | Full-training | 73.3 | 75.6 | 71.9 | 77.8 | 73.5 | 76.9 | – |
| | InfoBatch | 73.2 $^{-0.1}$ | 75.4 $^{-0.2}$ | 71.5 $^{-0.4}$ | **77.8** $^{+0.0}$ | **73.6** $^{+0.1}$ | 77.1 $^{+0.2}$ | -0.1 |
| | **InfoBatch + Ours** | **73.4** $^{+0.1}$ | **75.7** $^{+0.1}$ | **72.1** $^{+0.2}$ | **77.8** $^{+0.0}$ | **73.6** $^{+0.1}$ | **77.2** $^{+0.3}$ | **+0.1** |
| Noisy | Full-training | 66.4 | 67.3 | 64.7 | 68.9 | 65.0 | 67.8 | – |
| | InfoBatch | 65.8 $^{-0.6}$ | 66.8 $^{-0.5}$ | 64.1 $^{-0.6}$ | 68.1 $^{-0.8}$ | 64.3 $^{-0.7}$ | 66.9 $^{-0.9}$ | -0.7 |
| | **InfoBatch + Ours** | **67.2** $^{+0.8}$ | **67.8** $^{+0.5}$ | **65.5** $^{+0.8}$ | **69.4** $^{+0.5}$ | **65.6** $^{+0.6}$ | **68.2** $^{+0.4}$ | **+0.6** |

beyond ResNets. As shown in Table 4, AlignPrune consistently improves performance across various settings, confirming its ability to scale effectively to large-scale datasets and architectures.

## 4.4 ABLATION STUDIES

### 4.4.1 EFFECT OF HYPER-PARAMETERS

We conduct ablation experiments to investigate the effect of key hyper-parameters in AlignPrune, using CIFAR-100N with default settings across five noisy-label types and the clean-label case.

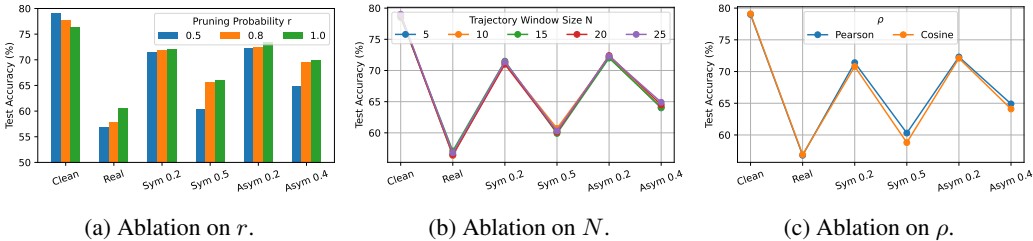

(a) Ablation on $r$.      (b) Ablation on $N$.      (c) Ablation on $\rho$.

Figure 2: **Ablation on hyper-parameter selection on CIFAR-100N.**

**Effect of Pruning Probability** $r$**.** We vary the pruning probability $r$ from 0.5 to 1.0 and evaluate performance under all six label conditions. As shown in Fig. 2a, increasing $r$ leads to decreased accuracy when trained on clean labels, while improves accuracy under all noisy-label types. This highlights that the proposed AlignPrune effectively identifies and discards noisy samples with a higher pruning probability, which also suggests that a lower $r$ should be used under clean-label case.

**Effect of Trajectory Window Size** $N$**.** We evaluate window sizes $N$ ranging from 5 to 25, to examine the sensitivity of the loss trajectory length. As shown in Fig. 2b, AlignPrune performs consistently across the range of $N$, indicating low sensitivity to the trajectory window size. In practice, we use $N = 25$, which corresponds to $12.5\%$ of the default 200 training epochs. We provide more detailed results on this ablation in Table G in Appendix B.2.2.

**Effect of Correlation Function** $\rho$**.** We compare two similarity functions used to compute the Dynamic Alignment Score: Pearson correlation and cosine similarity. As shown in Fig. 2c, both yield similar performance across all noise types. We adopt Pearson correlation by default due to its scale invariance and computational efficiency.

### 4.4.2 DEPENDENCE ON CLEAN DATA

To evaluate the robustness of AlignPrune in realistic scenarios where obtaining a large clean reference set is challenging, we conduct two additional ablation studies below with CIFAR-100N.

**Performance with Limited Clean Data.** We first assess the sensitivity of AlignPrune to the size of the clean reference set by varying its fraction from $100\%$ to $0.1\%$. This setup tests the method's ability to operate with minimal clean supervision.

Table 5: **Ablation on AlignPrune with estimated pseudo clean set.** We evaluate AlignPrune using pseudo clean subsets selected by SmallL (Jiang et al., 2018) and Moderate (Xia et al., 2022) as reference sets. A random subset from noisy training set is also included for comparison.

| Reference-set Fraction → | 100% | | 10% | | 1% | |
|---|---|---|---|---|---|---|
| Reference-set Type ↓ | Clean | Real | Clean | Real | Clean | Real |
| Noisy Train-set | 78.9 | 53.7 | 79.0 | 54.1 | 79.0 | 53.8 |
| Clean Validation-set | 79.0 | 56.8 | 78.9 | 56.3 | 79.1 | 56.0 |
| Pseudo Clean-set (estimated by Jiang et al. (2018)) | 79.1 | 56.0 | 79.1 | 56.8 | 79.0 | 56.4 |
| Pseudo Clean-set (estimated by Xia et al. (2022)) | 78.6 | 56.5 | 79.2 | 56.4 | 78.7 | 55.9 |

As shown in Fig. 3, AlignPrune maintains consistent performance across all six noisy-label types, as well as the clean-label condition, even with significantly reduced reference data scale. Remarkably, with only 10 clean samples ($0.1\%$ of the original clean set), the performance remains comparable to the full reference setting. A slight performance drop is observed for symmetric noise with noise rate of 0.8 under the $0.1\%$ reference scale. We attribute this to insufficient clean reference signal under extreme noisy condition. However, increasing the clean set fraction to just $0.2\%$ (doubling to 20 clean samples) can effectively mitigate this drop, showing the robustness of AlignPrune even under high-noisy scenarios.

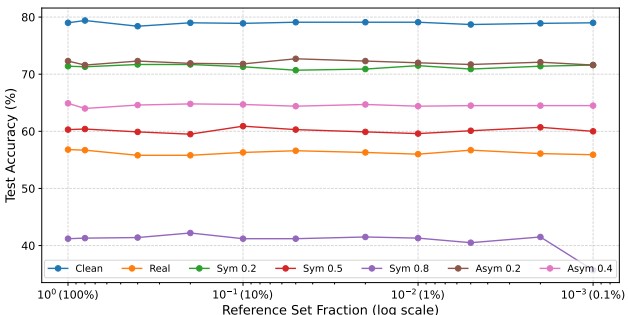

Figure 3: **Ablation on AlignPrune with varying reference set sizes.** Reference set size is reduced from $100\%$ to $0.1\%$ of the original one. AlignPrune remains effective even with minimal clean supervision. Best viewed in color.

**Performance with Pseudo Clean Data.** We further evaluate AlignPrune's dependency on having a clean reference set by replacing it with estimated pseudo-clean subsets selected directly from the noisy training data. Since most prior works do not aim to isolate purely clean samples, we use subsets generated by SmallL (Jiang et al., 2018) and Moderate (Xia et al., 2022), two static coreset methods known for identifying subsets with low noise levels (as reported in Table 4 of Park et al. (2023)).

Table 5 shows that using these pseudo-clean subsets yields comparable performance to using real clean reference sets, across both clean and noisy label conditions, even when the pseudo-clean subset is reduced to just $1\%$ of the reference set scale. We also use a random subset from the original noisy training set as the reference for comparison. Results show that while this performs similarly in the clean-label setting (where train and reference distributions match), it leads to significant accuracy drops under noisy conditions. This ablation confirms that AlignPrune can remain effective even when no explicit clean data is available, provided that a reasonably clean subset can be estimated **using noisy or even unlabeled data**, thus further highlighting its practical flexibility and robustness. We provide further discussions on the intuitions behind above ablation studies in Appendix B.2.3.

## 5 CONCLUSION

In this paper, we address the critical issue of label noise in dynamic data pruning, where existing loss-based methods often fail by mistakenly preserving high-loss noisy samples than the clean ones. To this end, we propose **AlignPrune**, a noise-robust and effective plug-and-play module that introduces a more reliable sample selection criterion: Dynamic Alignment Score (DAS). By leveraging the correlation between sample loss trajectories and a clean reference signal, AlignPrune effectively enables more accurate identification and pruning of noisy samples, which naturally generalizes to various label noise conditions. Extensive experiments on five benchmarks confirm the effectiveness of this proposed approach consistently across diverse real-world noisy-label conditions. Our results offer a generalizable solution for pruning under noisy data, encouraging future exploration of learning in real-world scenarios for broader applicability, especially in the era of large-scale models.

## ETHICS STATEMENT

Data pruning under label noise can improve model fairness and reliability by removing mislabeled or corrupted data, leading to more accurate and trustworthy AI systems. However, it also risks unintentionally discarding data from minority or underrepresented groups, potentially reinforcing existing biases. We therefore strongly recommend conducting task-specific validation and bias auditing when applying our results to real-world applications to ensure fair and inclusive outcomes.

## REPRODUCIBILITY STATEMENT

To ensure the reproducibility of our results, we commit to making our source code publicly available upon publication. The code will include our implementations and scripts to replicate the empirical results presented in this manuscript. Comprehensive implementation details of our proposed method, including dataset descriptions, experimental setups, noise injection details, and selection of baseline methods, are provided in Sec. 4.1 and further detailed in Appendix A. Additionally, all experiments are repeated three times to ensure the statistical reliability and stability. Table I provides the statistical significance analysis of our obtained results.

## LLM USAGE

We certify the use of LLMs to aid only in the writing and polishing of this manuscript.

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

CONTENTS

## A    ADDITIONAL IMPLEMENTATION DETAILS

We provide further implementation details to support reproducibility of our experimental setup below.

### A.1    DETAILS OF DATASET

Specifically, **CIFAR-100N** and **CIFAR-10N** (Wei et al., 2021) consist of 50K human re-annotated training images of size $32 \times 32$, with 100 and 10 classes respectively, adapted from the original CIFAR-100 and CIFAR-10 datasets (Krizhevsky et al., 2009). Both datasets include human-annotated noisy labels. We use the Real noisy-label set from CIFAR-100N, and both the Aggregate and Worst noisy-label sets from CIFAR-10N to represent *real* label noise. Following prior work (Mirzasoleiman et al., 2020b), we inject *synthetic* symmetric and asymmetric label noise with rates {0.2, 0.5, 0.8} and {0.2, 0.4} respectively, to simulate controllable noise. **WebVision** (Li et al., 2017) contains 2.4M images crawled from the Web, covering the same 1,000 categories as ImageNet-1K (Deng et al., 2009). Following standard practice in Chen et al. (2019), we use a subset of WebVision containing approximately 66K training images from the first 50 classes of the Google image. **Clothing-1M** (Xiao et al., 2015) consists of 1M training images sourced from online shopping platforms, with naturally occurring noisy labels. We follow standard practice and perform fine-tuning on the entire Clothing-1M training set using a pre-trained model weights on ImageNet. **ImageNet-1K** (Deng et al., 2009) consists of 1.2M training images with 1000 classes. We inject *synthetic* asymmetric label noise with rate 0.2 following settings in Park et al. (2023).

### A.2    DETAILS OF EXPERIMENTAL SETUP

Table A summarizes the experimental setups and hyper-parameters used across four main datasets and methods. For CIFAR-100N and CIFAR-10N, we follow the settings from Qin et al. (2024), using a ResNet-18 (He et al., 2016) backbone trained for 200 epochs optimized by LARS (You et al., 2019) with momentum of 0.9, weight decay of 5e-4, and batch size of 128. The initial learning rates are set to 5.2 (CIFAR-100N) and 5.62 (CIFAR-10N), both with a OneCycle learning rate scheduler (cosine annealing). For WebVision, we adopt the standard setup from Chen et al. (2019); Park et al. (2023), using an InceptionResNetV2 (Szegedy et al., 2017) trained for 100 epochs optimized by SGD with batch size of 32. For Clothing-1M, we fine-tune a ResNet-50 (He et al., 2016) pre-trained on ImageNet-1K (Deng et al., 2009) for 10 epochs using a batch size of 32. The initial learning rates of WebVision and Clothing-1M are set to 0.02 and 0.002, decayed by a factor of 10 halfway through the total training epochs. For ImageNet-1K, we use the default settings for corresponding architectures with 100 epoch training. Unless otherwise stated, we apply standard data augmentations, including normalization, random cropping, and horizontal flipping for all images during model training.

To ensure fairness and reproducibility, we adopt the default optimal hyper-parameter settings from the original papers for both static and dynamic data pruning baselines. For InfoBatch (Qin et al., 2024), we use the pruning probability $r = 0.5$ for CIFAR-100N and CIFAR-10N, and a more aggressive $r = 0.75$ for the large-scale datasets WebVision and Clothing-1M. For SeTa (Zhou et al., 2025), we use $r = 0.1$ with window scale $\alpha = 0.9$ and group number $k = 5$ on CIFAR-100N and CIFAR-10N. Due to training collapse of SeTa on WebVision and Clothing-1M, we only report results with InfoBatch on these two datasets. The annealing ratio $\delta$ is set to 0.875 for both InfoBatch and SeTa across all settings. For our method *AlignPrune*, we follow the same pruning hyper-parameters as the corresponding dynamic pruning method. The trajectory window size $N$ is set to 25 by default, and reduced to 5 for Clothing-1M due to its short fine-tuning schedule. Pearson's correlation is used as the default correlation function $\rho$.

### A.3    DETAILS OF NOISE INJECTION

For *synthetic* label noise on CIFAR-100N and CIFAR-10N, we follow the protocol in Mirzasoleiman et al. (2020b). Specifically, we inject: **Symmetric noise** with rates {0.2, 0.5, 0.8}, where $r\%$ of the labels from each class $c$ are randomly flipped to the next consecutive class $c + 1$, and labels from the final class are wrapped around to class 0. **Asymmetric noise** with rates {0.2, 0.4}, where selected samples are flipped within their super-class as defined in Wei et al. (2021). For ImageNet-1K experiments, we follow Park et al. (2023) to inject asymmetric noise with rate of 0.2. This controlled

Table A: **Summary of the experimental setups and hyper-parameters on CIFAR-100N, CIFAR-10N, WebVision and Clothing-1M datasets.**

| Hyper-paramters | | CIFAR-100N | CIFAR-10N | WebVision | Clothing-1M |
|---|---|---|---|---|---|
| **Training Setup** | architecture | ResNet-18 | ResNet-18 | InceptionResNetV2 | ResNet-50 (Pre-trained) |
| | training epoch | 200 | 200 | 100 | 10 |
| | batch size | 128 | 128 | 32 | 32 |
| | optimizer | LARS | LARS | SGD | SGD |
| | learning rate | 5.2 | 5.62 | 0.02 | 0.002 |
| | lr scheduler | OneCycle | OneCycle | MultiStep-50th | MultiStep-5th |
| | weight decay | $5 \times 10^{-4}$ | $5 \times 10^{-4}$ | $5 \times 10^{-4}$ | $5 \times 10^{-4}$ |
| **InfoBatch** | $r$ | 0.5 | 0.5 | 0.75 | 0.75 |
| | $\delta$ | 0.875 | 0.875 | 0.875 | 0.875 |
| **SeTa** | $r$ | 0.1 | 0.1 | | |
| | $\alpha$ | 0.9 | 0.9 | – | – |
| | $k$ | 5 | 5 | | |
| | $\delta$ | 0.875 | 0.875 | | |
| **Ours** | $N$ | 25 | 25 | 25 | 5 |
| | $\rho$ | Pearson | Pearson | Pearson | Pearson |

noise injection allows us to benchmark the robustness of different baselines across varying noisy types and noisy rates.

### A.4 DETAILS OF BASELINE METHOD

We clarifiy the selection criteria for dynamic pruning baselines as follows: Several recent methods are excluded from our evaluation due to incompatibility with our experimental setting. Specifically, SCAN (Guo & Kankanhalli, 2024) and DISSect (Zhao et al., 2025) are tailored for contrastive pre-training frameworks like CLIP (Radford et al., 2021), which significantly differ from our focus on noisy learning. Additionally, methods by Raju et al. (2021), Yang et al. (2025) and Wu et al. (2025) are excluded due to the lack of publicly available implementations, and their reported results are limited to clean-label scenarios with different hyper-parameters, making fair and reproducible comparison infeasible.

**Remark.** Although AlignPrune is designed as a plug-and-play replacement to enhance dynamic pruning methods, its integration assumes an epoch-level sample-wise scoring and ranking mechanism based on loss trajectories. We selected InfoBatch (Qin et al., 2024) and SeTa (Zhou et al., 2025) as representative dynamic pruning baselines because both provide public implementations that align well with our framework. In contrast, DivBS (Hong et al., 2024) and IES (Yuan et al., 2025) adopt fundamentally different designs: DivBS performs pruning at batch level, making per-sample loss trajectory inapplicable. Moreover, applying such batch-level pruning under label noise tends to remove meaningful samples while retaining noisy ones. IES, while structurally compatible, becomes functionally equivalent to InfoBatch once AlignPrune is applied, with the only difference lies in the use of a fixed threshold versus a dynamic mean-based threshold. Therefore, we integrate AlignPrune only with InfoBatch and SeTa, while including DivBS and IES in the comparison for completeness.

## B ADDITIONAL EXPERIMENTAL RESULTS

### B.1 EXTENDED BASELINE COMPARISONS

**Main Results under Label Noise.** Tables B and C present a comprehensive comparison of Align-Prune with both static and dynamic data pruning baselines on CIFAR-100N and CIFAR-10N under three pruning ratios. Overall, static pruning methods perform significantly worse than dynamic methods, particularly in the presence of label noise. This highlights the limitations of static coreset selection when noisy labels are involved. In contrast, the proposed AlignPrune consistently achieves

Table B: **Classification results on CIFAR-100N with ResNet-18.** Performance gaps relative to the full-training setting are indicated as superscripts. Static pruning baselines are highlighted with gray. The mean $\Delta$ is computed across all noise types. $^\dagger$ indicates Prune4ReL with standard CE loss to ensure fair comparison on pure pruning methods.

| Noisy Type → | Clean | Real | Symmetric | | | Asymmetric | | Mean |
| Pruning Method ↓ | | | 0.2 | 0.5 | 0.8 | 0.2 | 0.4 | $\Delta$ |
|---|---|---|---|---|---|---|---|---|
| Full-training | 78.2 | 56.1 | 71.4 | 58.6 | 39.8 | 72.4 | 63.3 | – |
| **Prune Ratio ~30%** | | | | | | | | |
| Static Random | $73.8^{-4.4}$ | $53.3^{-2.8}$ | $67.9^{-3.4}$ | $51.1^{-7.5}$ | $30.3^{-9.5}$ | $68.4^{-4.0}$ | $57.8^{-5.5}$ | -5.3 |
| SmallL (Jiang et al., 2018) | $71.3^{-6.9}$ | $58.0^{+1.9}$ | $69.2^{-2.2}$ | $51.9^{-6.7}$ | $13.7^{-26.1}$ | $68.2^{-4.2}$ | $57.0^{-6.2}$ | -7.2 |
| Margin (Coleman et al., 2019) | $75.4^{-2.8}$ | $48.0^{-8.1}$ | $59.2^{-12.2}$ | $15.9^{-42.7}$ | $6.3^{-33.5}$ | $57.3^{-15.1}$ | $42.1^{-21.2}$ | -19.4 |
| Forget (Toneva et al., 2018) | $74.8^{-3.4}$ | $57.7^{+1.6}$ | $70.2^{-1.1}$ | $51.3^{-7.3}$ | $12.5^{-27.4}$ | $68.4^{-4.0}$ | $59.2^{-4.1}$ | -6.5 |
| GraNd (Paul et al., 2021) | $73.0^{-5.2}$ | $40.2^{-15.9}$ | $49.4^{-21.9}$ | $9.3^{-49.3}$ | $6.2^{-33.6}$ | $51.8^{-20.5}$ | $30.3^{-33.0}$ | -25.6 |
| Moderate (Xia et al., 2022) | $73.2^{-5.0}$ | $52.1^{-4.0}$ | $60.2^{-11.2}$ | $30.2^{-28.4}$ | $9.9^{-29.9}$ | $60.0^{-12.4}$ | $45.7^{-17.6}$ | -15.5 |
| SSP (Sorscher et al., 2022) | $74.9^{-3.3}$ | $53.0^{-3.1}$ | $60.3^{-11.1}$ | $35.8^{-22.9}$ | $7.9^{-31.9}$ | $62.9^{-9.5}$ | $50.4^{-12.8}$ | -13.5 |
| Prune4ReL$^\dagger$ (Park et al., 2023) | $73.2^{-5.0}$ | $52.3^{-3.8}$ | $61.2^{-10.1}$ | $37.0^{-21.6}$ | $9.0^{-30.8}$ | $61.1^{-11.3}$ | $48.9^{-14.4}$ | -13.8 |
| Dyn-Unc (He et al., 2024) | $77.2^{-1.0}$ | $58.5^{+2.3}$ | $70.9^{-0.5}$ | $55.4^{-3.2}$ | $24.3^{-15.5}$ | $71.8^{-0.6}$ | $62.6^{-0.7}$ | -2.7 |
| DUAL (Cho et al., 2025) | $78.1^{-0.1}$ | $58.8^{+2.7}$ | $71.3^{-0.1}$ | $55.4^{-3.2}$ | $22.7^{-17.1}$ | $72.2^{-0.2}$ | $60.8^{-2.4}$ | -2.9 |
| Dynamic Random | $77.3^{-0.9}$ | $54.7^{-1.4}$ | $69.9^{-1.4}$ | $58.8^{+0.2}$ | $40.1^{+0.3}$ | $71.5^{-0.9}$ | $63.2^{-0.1}$ | -0.6 |
| DivBS (Hong et al., 2024) | $77.1^{-1.1}$ | $54.6^{-1.5}$ | $63.5^{-7.9}$ | $51.0^{-7.6}$ | $20.8^{-19.0}$ | $65.9^{-6.5}$ | $56.2^{-7.1}$ | -7.2 |
| IES (Yuan et al., 2025) | $76.1^{-2.1}$ | $53.9^{-2.2}$ | $64.2^{-7.2}$ | $51.0^{-7.6}$ | $37.0^{-2.8}$ | $65.3^{-7.0}$ | $53.8^{-9.5}$ | -5.5 |
| InfoBatch (Qin et al., 2024) | $79.0^{+0.8}$ | $56.1^{+0.0}$ | $71.4^{+0.0}$ | $59.7^{+1.1}$ | $41.8^{+2.0}$ | $71.9^{-0.5}$ | $64.2^{+0.9}$ | +0.6 |
| **InfoBatch + Ours** | $79.3^{+1.1}$ | $59.4^{+3.3}$ | $71.8^{+0.4}$ | $66.0^{+7.4}$ | $41.8^{+2.0}$ | $72.6^{+0.2}$ | $68.0^{+4.7}$ | **+2.7** |
| SeTa (Zhou et al., 2025) | $79.0^{+0.8}$ | $55.6^{-0.5}$ | $70.2^{-1.2}$ | $59.0^{+0.4}$ | $41.6^{+1.8}$ | $71.4^{-0.9}$ | $63.2^{-0.1}$ | +0.0 |
| **SeTa + Ours** | $79.3^{+1.1}$ | $56.3^{+0.2}$ | $70.8^{-0.5}$ | $60.5^{+1.9}$ | $41.6^{+1.8}$ | $71.9^{-0.5}$ | $64.3^{+1.0}$ | **+0.7** |
| **Prune Ratio ~50%** | | | | | | | | |
| Static Random | $72.1^{-6.1}$ | $51.8^{-4.3}$ | $64.0^{-7.4}$ | $47.2^{-11.5}$ | $22.9^{-16.9}$ | $65.3^{-7.0}$ | $54.6^{-8.7}$ | -8.8 |
| SmallL (Jiang et al., 2018) | $65.8^{-12.4}$ | $56.1^{-0.1}$ | $64.9^{-6.4}$ | $57.9^{-0.8}$ | $18.0^{-21.8}$ | $64.2^{-8.1}$ | $58.8^{-4.4}$ | -7.7 |
| Margin (Coleman et al., 2019) | $71.2^{-7.0}$ | $36.3^{-19.8}$ | $44.7^{-26.7}$ | $11.0^{-47.6}$ | $5.9^{-33.9}$ | $43.7^{-28.7}$ | $26.8^{-36.5}$ | -28.6 |
| Forget (Toneva et al., 2018) | $69.8^{-8.4}$ | $56.4^{+0.3}$ | $68.4^{-2.9}$ | $60.4^{+1.8}$ | $16.5^{-23.3}$ | $66.1^{-6.3}$ | $60.7^{-2.5}$ | -5.9 |
| GraNd (Paul et al., 2021) | $64.0^{-14.2}$ | $25.8^{-30.3}$ | $28.6^{-42.8}$ | $4.1^{-54.5}$ | $4.3^{-35.5}$ | $36.6^{-35.8}$ | $18.2^{-45.1}$ | -36.9 |
| Moderate (Xia et al., 2022) | $68.7^{-9.5}$ | $49.0^{-7.2}$ | $52.7^{-18.6}$ | $22.3^{-36.3}$ | $6.4^{-33.4}$ | $55.1^{-17.3}$ | $41.0^{-22.3}$ | -20.6 |
| SSP (Sorscher et al., 2022) | $70.6^{-7.6}$ | $49.2^{-6.9}$ | $52.9^{-18.5}$ | $31.1^{-27.5}$ | $7.5^{-32.3}$ | $58.6^{-13.8}$ | $47.5^{-15.8}$ | -17.5 |
| Prune4ReL$^\dagger$ (Park et al., 2023) | $69.1^{-9.1}$ | $51.7^{-4.4}$ | $58.2^{-13.2}$ | $32.7^{-25.9}$ | $7.4^{-32.4}$ | $58.7^{-13.7}$ | $45.3^{-17.9}$ | -16.7 |
| Dyn-Unc (He et al., 2024) | $74.1^{-4.2}$ | $59.7^{+3.5}$ | $71.4^{+0.0}$ | $61.7^{+3.1}$ | $22.8^{-17.0}$ | $71.1^{-1.2}$ | $66.5^{+3.2}$ | -1.8 |
| DUAL (Cho et al., 2025) | $74.6^{-3.6}$ | $59.9^{+3.8}$ | $71.3^{-0.1}$ | $61.9^{+3.3}$ | $20.6^{-19.2}$ | $71.1^{-1.3}$ | $62.9^{-0.4}$ | -2.5 |
| Dynamic Random | $75.3^{-2.9}$ | $54.1^{-2.0}$ | $70.4^{-1.0}$ | $59.5^{+0.9}$ | $40.7^{+0.9}$ | $70.1^{-2.3}$ | $64.8^{+1.6}$ | -0.7 |
| DivBS (Hong et al., 2024) | $77.1^{-1.1}$ | $53.8^{-2.3}$ | $63.8^{-7.6}$ | $46.6^{-12.0}$ | $18.0^{-21.9}$ | $64.5^{-7.8}$ | $56.7^{-6.6}$ | -8.5 |
| IES (Yuan et al., 2025) | $75.5^{-2.7}$ | $53.2^{-2.9}$ | $62.7^{-8.7}$ | $50.2^{-8.4}$ | $34.4^{-5.4}$ | $63.3^{-9.1}$ | $54.8^{-8.5}$ | -6.5 |
| InfoBatch (Qin et al., 2024) | $77.7^{-0.5}$ | $56.0^{-0.1}$ | $71.3^{-0.1}$ | $60.5^{+1.9}$ | $42.2^{+2.4}$ | $71.8^{-0.6}$ | $65.2^{+2.0}$ | +0.7 |
| **InfoBatch + Ours** | $78.5^{+0.3}$ | $60.7^{+4.6}$ | $71.6^{+0.3}$ | $62.0^{+3.4}$ | $42.6^{+2.8}$ | $72.6^{+0.3}$ | $68.6^{+5.3}$ | **+2.4** |
| SeTa (Zhou et al., 2025) | $77.5^{-0.7}$ | $55.7^{-0.4}$ | $70.7^{-0.7}$ | $60.0^{+1.4}$ | $40.5^{+0.7}$ | $71.9^{-0.5}$ | $64.5^{+1.2}$ | +0.2 |
| **SeTa + Ours** | $78.4^{+0.2}$ | $56.3^{+0.1}$ | $71.2^{-0.2}$ | $61.0^{+2.4}$ | $41.9^{+2.1}$ | $72.2^{-0.2}$ | $66.0^{+2.7}$ | **+1.0** |
| **Prune Ratio ~70%** | | | | | | | | |
| Static Random | $69.7^{-8.5}$ | $45.9^{-10.2}$ | $56.7^{-14.6}$ | $39.0^{-19.6}$ | $15.8^{-24.0}$ | $58.4^{-13.9}$ | $49.5^{-13.8}$ | -14.9 |
| SmallL (Jiang et al., 2018) | $55.7^{-22.5}$ | $49.8^{-6.3}$ | $57.0^{-14.4}$ | $54.6^{-4.1}$ | $22.0^{-17.8}$ | $56.7^{-15.7}$ | $53.8^{-9.5}$ | -12.9 |
| Margin (Coleman et al., 2019) | $56.7^{-21.5}$ | $10.6^{-45.5}$ | $12.3^{-59.1}$ | $6.5^{-52.2}$ | $4.5^{-35.3}$ | $17.1^{-55.3}$ | $5.2^{-58.1}$ | -46.7 |
| Forget (Toneva et al., 2018) | $58.6^{-19.6}$ | $50.0^{-6.1}$ | $61.8^{-9.6}$ | $59.3^{+0.7}$ | $22.8^{-17.0}$ | $58.4^{-14.0}$ | $56.1^{-7.2}$ | -10.4 |
| GraNd (Paul et al., 2021) | $46.2^{-32.0}$ | $14.7^{-41.4}$ | $10.1^{-61.3}$ | $2.9^{-55.8}$ | $3.4^{-36.4}$ | $20.1^{-52.3}$ | $9.1^{-54.2}$ | -47.6 |
| Moderate (Xia et al., 2022) | $60.6^{-17.6}$ | $43.3^{-12.8}$ | $46.2^{-25.2}$ | $17.7^{-41.0}$ | $5.1^{-34.7}$ | $46.6^{-25.8}$ | $36.5^{-26.7}$ | -26.3 |
| SSP (Sorscher et al., 2022) | $60.3^{-17.9}$ | $42.2^{-14.0}$ | $41.3^{-30.1}$ | $21.6^{-37.0}$ | $6.1^{-33.8}$ | $49.2^{-23.2}$ | $37.7^{-25.6}$ | -25.9 |
| Prune4ReL$^\dagger$ (Park et al., 2023) | $61.4^{-16.8}$ | $46.2^{-10.0}$ | $51.1^{-20.3}$ | $28.0^{-30.7}$ | $6.1^{-33.7}$ | $52.4^{-20.0}$ | $42.8^{-20.5}$ | -21.7 |
| Dyn-Unc (He et al., 2024) | $63.9^{-14.3}$ | $57.5^{+1.4}$ | $67.1^{-4.3}$ | $63.4^{+4.7}$ | $24.6^{-15.2}$ | $65.8^{-6.6}$ | $63.0^{-0.2}$ | -4.9 |
| DUAL (Cho et al., 2025) | $68.9^{-9.3}$ | $57.9^{+1.7}$ | $66.4^{-5.0}$ | $62.5^{+3.9}$ | $19.9^{-19.9}$ | $67.3^{-5.1}$ | $61.5^{-1.8}$ | -5.1 |
| Dynamic Random | $75.2^{-3.0}$ | $53.3^{-2.9}$ | $70.2^{-1.2}$ | $62.7^{+4.1}$ | $41.0^{+1.2}$ | $71.9^{-0.5}$ | $67.3^{+4.0}$ | +0.3 |
| DivBS (Hong et al., 2024) | $76.9^{-1.3}$ | $53.6^{-2.6}$ | $61.5^{-9.8}$ | $45.8^{-12.5}$ | $19.4^{-20.4}$ | $63.5^{-8.8}$ | $58.5^{-4.8}$ | -8.6 |
| IES (Yuan et al., 2025) | $74.8^{-3.4}$ | $52.6^{-3.6}$ | $61.0^{-10.4}$ | $50.3^{-8.3}$ | $28.6^{-11.2}$ | $62.6^{-9.8}$ | $51.9^{-11.4}$ | -8.3 |
| InfoBatch (Qin et al., 2024) | $77.1^{-1.1}$ | $55.0^{-1.2}$ | $71.4^{+0.1}$ | $63.6^{+5.0}$ | $42.4^{+2.6}$ | $72.2^{-0.2}$ | $67.8^{+4.5}$ | +1.4 |
| **InfoBatch + Ours** | $77.5^{-0.7}$ | $58.3^{+2.2}$ | $72.2^{+0.9}$ | $64.7^{+6.1}$ | $42.8^{+3.0}$ | $72.5^{+0.1}$ | $69.1^{+5.9}$ | **+2.5** |
| SeTa (Zhou et al., 2025) | $77.2^{-1.1}$ | $55.2^{-1.0}$ | $71.6^{+0.2}$ | $62.4^{+3.8}$ | $42.4^{+2.6}$ | $72.0^{-0.4}$ | $67.4^{+4.1}$ | +1.2 |
| **SeTa + Ours** | $77.8^{-0.4}$ | $55.5^{-0.6}$ | $72.3^{+0.9}$ | $63.3^{+4.7}$ | $42.7^{+2.9}$ | $72.7^{+0.3}$ | $67.6^{+4.3}$ | **+1.7** |

Table C: **Classification results on CIFAR-10N with ResNet-18.** Performance gaps relative to the full-training setting are indicated as superscripts. Static pruning methods are highlighted with gray. The mean $\Delta$ is computed across all noise types. $^\dagger$ indicates Prune4ReL with standard CE loss to ensure fair comparison on pure pruning methods.

| Noisy Type → | Clean | Real | | Symmetric | | | Asymmetric | | Mean |
|---|---|---|---|---|---|---|---|---|---|
| Pruning Method ↓ | | Real-A | Real-W | 0.2 | 0.5 | 0.8 | 0.2 | 0.4 | $\Delta$ |
| Full-training | 95.6 | 90.7 | 78.3 | 91.3 | 85.4 | 65.6 | 90.6 | 85.8 | – |
| **Prune Ratio $\sim$30%** | | | | | | | | | |
| Static Random | 94.6$^{-1.0}$ | 89.9$^{-0.7}$ | 75.0$^{-3.3}$ | 89.0$^{-2.3}$ | 81.2$^{-4.2}$ | 57.6$^{-8.0}$ | 87.9$^{-2.8}$ | 82.5$^{-3.3}$ | -3.2 |
| SmallL (Jiang et al., 2018) | 88.9$^{-6.7}$ | 87.2$^{-3.5}$ | 78.2$^{-0.1}$ | 89.3$^{-2.0}$ | 72.1$^{-13.3}$ | 24.0$^{-41.6}$ | 87.4$^{-3.3}$ | 68.9$^{-16.9}$ | -10.9 |
| Margin (Coleman et al., 2019) | 94.5$^{-1.1}$ | 88.9$^{-1.8}$ | 64.6$^{-13.7}$ | 85.4$^{-5.9}$ | 15.0$^{-70.5}$ | 12.0$^{-53.6}$ | 79.8$^{-10.8}$ | 47.6$^{-38.2}$ | -24.4 |
| Forget (Toneva et al., 2018) | 93.6$^{-2.0}$ | 89.5$^{-1.1}$ | 69.6$^{-8.8}$ | 84.5$^{-6.8}$ | 59.0$^{-26.4}$ | 22.5$^{-43.1}$ | 80.5$^{-10.2}$ | 58.3$^{-27.5}$ | -15.7 |
| GraNd (Paul et al., 2021) | 93.9$^{-1.7}$ | 87.8$^{-2.9}$ | 40.5$^{-37.8}$ | 65.9$^{-25.4}$ | 17.5$^{-68.0}$ | 11.0$^{-54.6}$ | 60.5$^{-30.1}$ | 38.7$^{-47.1}$ | -33.4 |
| Moderate (Xia et al., 2022) | 92.1$^{-3.5}$ | 89.9$^{-0.7}$ | 66.8$^{-11.5}$ | 87.8$^{-3.5}$ | 52.8$^{-32.6}$ | 17.8$^{-47.9}$ | 80.8$^{-9.9}$ | 58.3$^{-27.5}$ | -17.1 |
| SSP (Sorscher et al., 2022) | 93.4$^{-2.2}$ | 88.6$^{-2.1}$ | 66.0$^{-12.3}$ | 83.4$^{-7.9}$ | 53.1$^{-32.4}$ | 16.6$^{-49.0}$ | 80.3$^{-10.4}$ | 59.5$^{-26.3}$ | -17.8 |
| Prune4ReL$^\dagger$ (Park et al., 2023) | 93.0$^{-2.6}$ | 88.3$^{-2.4}$ | 66.1$^{-12.3}$ | 81.8$^{-9.5}$ | 52.2$^{-33.2}$ | 15.8$^{-49.8}$ | 81.1$^{-9.6}$ | 59.0$^{-26.8}$ | -18.3 |
| Dyn-Unc (He et al., 2024) | 95.3$^{-0.3}$ | 89.6$^{-1.0}$ | 78.7$^{+0.4}$ | 89.6$^{-1.7}$ | 78.6$^{-6.9}$ | 44.8$^{-20.8}$ | 82.6$^{-8.0}$ | 76.1$^{-9.7}$ | -6.0 |
| DUAL (Cho et al., 2025) | 95.5$^{-0.1}$ | 90.1$^{-0.5}$ | 80.4$^{+2.0}$ | 91.7$^{+0.4}$ | 79.2$^{-6.2}$ | 40.2$^{-25.5}$ | 87.1$^{-3.6}$ | 67.8$^{-18.0}$ | -6.4 |
| Dynamic Random | 94.8$^{-0.8}$ | 90.3$^{-0.3}$ | 77.3$^{-1.0}$ | 90.8$^{-0.5}$ | 84.2$^{-1.2}$ | 64.8$^{-0.8}$ | 91.5$^{+0.9}$ | 84.5$^{-1.3}$ | -0.6 |
| DivBS (Hong et al., 2024) | 95.3$^{-0.3}$ | 90.3$^{-0.3}$ | 78.3$^{+0.0}$ | 85.3$^{-6.0}$ | 79.2$^{-6.3}$ | 50.3$^{-15.3}$ | 88.0$^{-2.6}$ | 82.8$^{-3.0}$ | -4.2 |
| IES (Yuan et al., 2025) | 95.0$^{-0.6}$ | 89.8$^{-0.8}$ | 79.8$^{+1.5}$ | 87.5$^{-3.8}$ | 80.0$^{-5.4}$ | 46.0$^{-19.7}$ | 88.9$^{-1.8}$ | 82.3$^{-3.5}$ | -4.3 |
| InfoBatch (Qin et al., 2024) | 95.3$^{-0.3}$ | 91.0$^{+0.3}$ | 78.1$^{-0.3}$ | 92.0$^{+0.7}$ | 85.6$^{+0.1}$ | 67.5$^{+1.9}$ | 91.3$^{+0.7}$ | 87.0$^{+1.3}$ | +0.5 |
| **InfoBatch + Ours** | 95.3$^{-0.3}$ | 91.4$^{+0.7}$ | 82.0$^{+3.7}$ | 92.7$^{+1.4}$ | 87.9$^{+2.5}$ | 67.8$^{+2.2}$ | 92.9$^{+2.3}$ | 90.0$^{+4.2}$ | **+2.1** |
| SeTa (Zhou et al., 2025) | 95.2$^{-0.4}$ | 90.6$^{-0.1}$ | 76.3$^{-2.0}$ | 91.8$^{+0.5}$ | 84.7$^{-0.7}$ | 64.2$^{-1.5}$ | 90.8$^{+0.1}$ | 88.4$^{+2.6}$ | -0.2 |
| **SeTa + Ours** | 95.3$^{-0.3}$ | 90.7$^{+0.0}$ | 79.1$^{+0.8}$ | 92.5$^{+1.2}$ | 85.0$^{-0.5}$ | 64.6$^{-1.0}$ | 91.8$^{+1.2}$ | 89.3$^{+3.5}$ | **+0.6** |
| **Prune Ratio $\sim$50%** | | | | | | | | | |
| Static Random | 93.3$^{-2.3}$ | 88.6$^{-2.1}$ | 72.4$^{-5.9}$ | 87.4$^{-3.9}$ | 77.5$^{-8.0}$ | 51.4$^{-14.2}$ | 85.8$^{-4.9}$ | 77.9$^{-7.9}$ | -6.1 |
| SmallL (Jiang et al., 2018) | 84.1$^{-11.5}$ | 84.1$^{-6.6}$ | 78.0$^{-0.3}$ | 84.7$^{-6.6}$ | 80.8$^{-4.6}$ | 30.9$^{-34.7}$ | 84.7$^{-6.0}$ | 60.9$^{-24.9}$ | -11.9 |
| Margin (Coleman et al., 2019) | 93.3$^{-2.3}$ | 85.2$^{-5.5}$ | 44.6$^{-33.8}$ | 70.7$^{-20.7}$ | 13.5$^{-72.0}$ | 10.7$^{-54.9}$ | 57.5$^{-33.1}$ | 38.6$^{-47.2}$ | -33.7 |
| Forget (Toneva et al., 2018) | 92.4$^{-3.2}$ | 88.4$^{-2.3}$ | 72.2$^{-6.1}$ | 85.7$^{-5.6}$ | 68.3$^{-17.2}$ | 29.8$^{-35.8}$ | 79.9$^{-10.7}$ | 57.4$^{-28.4}$ | -13.7 |
| GraNd (Paul et al., 2021) | 91.2$^{-4.4}$ | 78.6$^{-12.0}$ | 21.0$^{-57.4}$ | 39.2$^{-52.1}$ | 7.1$^{-78.4}$ | 6.9$^{-58.7}$ | 37.2$^{-53.5}$ | 32.3$^{-53.5}$ | -46.2 |
| Moderate (Xia et al., 2022) | 90.8$^{-4.8}$ | 88.2$^{-2.4}$ | 66.9$^{-11.4}$ | 89.9$^{-1.4}$ | 48.8$^{-36.6}$ | 17.4$^{-48.2}$ | 80.1$^{-10.5}$ | 57.7$^{-28.1}$ | -17.9 |
| SSP (Sorscher et al., 2022) | 91.5$^{-4.1}$ | 87.1$^{-3.6}$ | 65.8$^{-12.6}$ | 81.2$^{-10.1}$ | 50.7$^{-34.8}$ | 16.3$^{-49.4}$ | 79.4$^{-11.2}$ | 58.6$^{-27.2}$ | -19.1 |
| Prune4ReL$^\dagger$ (Park et al., 2023) | 91.6$^{-4.1}$ | 85.9$^{-4.8}$ | 63.9$^{-14.5}$ | 78.4$^{-12.9}$ | 49.6$^{-35.9}$ | 16.3$^{-49.3}$ | 78.8$^{-11.8}$ | 56.7$^{-29.1}$ | -20.3 |
| Dyn-Unc (He et al., 2024) | 95.4$^{-0.2}$ | 89.3$^{-1.4}$ | 78.8$^{+0.5}$ | 90.9$^{-0.4}$ | 84.5$^{-1.0}$ | 47.1$^{-18.5}$ | 80.7$^{-10.0}$ | 71.2$^{-14.6}$ | -5.7 |
| DUAL (Cho et al., 2025) | 95.4$^{-0.2}$ | 90.1$^{-0.6}$ | 81.3$^{+3.4}$ | 92.0$^{+0.7}$ | 83.6$^{-1.8}$ | 41.8$^{-23.8}$ | 89.0$^{-1.7}$ | 75.3$^{-10.5}$ | -4.3 |
| Dynamic Random | 94.5$^{-1.1}$ | 89.2$^{-1.5}$ | 76.1$^{-2.3}$ | 88.9$^{-2.4}$ | 83.7$^{-1.8}$ | 63.3$^{-2.3}$ | 90.6$^{+0.0}$ | 83.7$^{-2.1}$ | -1.7 |
| DivBS (Hong et al., 2024) | 94.9$^{-0.7}$ | 89.8$^{-0.8}$ | 78.0$^{-0.4}$ | 85.9$^{-5.4}$ | 79.6$^{-5.9}$ | 48.6$^{-17.0}$ | 88.8$^{-1.8}$ | 84.6$^{-1.2}$ | -4.2 |
| IES (Yuan et al., 2025) | 95.0$^{-0.6}$ | 89.8$^{-0.8}$ | 78.6$^{+0.2}$ | 87.0$^{-4.3}$ | 79.1$^{-6.4}$ | 35.7$^{-29.9}$ | 88.5$^{-2.2}$ | 78.3$^{-7.5}$ | -6.4 |
| InfoBatch (Qin et al., 2024) | 95.0$^{-0.6}$ | 90.5$^{-0.2}$ | 77.7$^{-0.6}$ | 92.0$^{+0.7}$ | 87.5$^{+2.1}$ | 64.3$^{-1.3}$ | 92.0$^{+1.3}$ | 87.5$^{+1.7}$ | +0.4 |
| **InfoBatch + Ours** | 95.0$^{-0.6}$ | 91.3$^{+0.6}$ | 82.4$^{+4.1}$ | 92.8$^{+1.5}$ | 87.6$^{+2.1}$ | 64.5$^{-1.1}$ | 92.1$^{+1.5}$ | 89.8$^{+4.0}$ | **+1.5** |
| SeTa (Zhou et al., 2025) | 94.8$^{-0.8}$ | 89.4$^{-1.3}$ | 78.5$^{+0.2}$ | 91.3$^{+0.0}$ | 86.0$^{+0.5}$ | 53.0$^{-12.7}$ | 91.7$^{+1.1}$ | 87.9$^{+2.1}$ | -1.4 |
| **SeTa + Ours** | 94.9$^{-0.7}$ | 89.7$^{-1.0}$ | 79.3$^{+1.0}$ | 91.9$^{+0.6}$ | 87.6$^{+2.2}$ | 60.2$^{-5.5}$ | 92.2$^{+1.6}$ | 88.5$^{+2.7}$ | **+0.1** |
| **Prune Ratio $\sim$70%** | | | | | | | | | |
| Static Random | 90.2$^{-5.4}$ | 86.6$^{-4.1}$ | 69.0$^{-9.3}$ | 85.3$^{-6.0}$ | 73.8$^{-11.6}$ | 41.4$^{-24.3}$ | 82.3$^{-8.3}$ | 75.3$^{-10.5}$ | -9.9 |
| SmallL (Jiang et al., 2018) | 75.1$^{-20.5}$ | 76.3$^{-14.3}$ | 74.4$^{-4.0}$ | 78.2$^{-13.1}$ | 76.1$^{-9.4}$ | 35.4$^{-30.2}$ | 73.1$^{-17.6}$ | 62.7$^{-23.1}$ | -16.5 |
| Margin (Coleman et al., 2019) | 91.2$^{-4.4}$ | 71.7$^{-19.0}$ | 15.1$^{-63.3}$ | 22.6$^{-68.7}$ | 12.3$^{-73.1}$ | 9.8$^{-55.8}$ | 34.0$^{-56.7}$ | 27.3$^{-58.5}$ | -49.9 |
| Forget (Toneva et al., 2018) | 89.8$^{-5.8}$ | 86.2$^{-4.5}$ | 70.9$^{-7.5}$ | 85.5$^{-5.8}$ | 71.5$^{-13.9}$ | 33.0$^{-32.7}$ | 78.4$^{-12.2}$ | 55.2$^{-30.6}$ | -14.1 |
| GraNd (Paul et al., 2021) | 78.1$^{-17.5}$ | 52.1$^{-38.6}$ | 10.2$^{-68.1}$ | 14.3$^{-77.0}$ | 8.2$^{-77.2}$ | 7.8$^{-57.8}$ | 18.9$^{-71.8}$ | 28.5$^{-57.3}$ | -58.1 |
| Moderate (Xia et al., 2022) | 87.8$^{-7.8}$ | 84.8$^{-5.8}$ | 66.3$^{-12.0}$ | 87.3$^{-4.0}$ | 47.8$^{-37.7}$ | 16.7$^{-48.9}$ | 77.7$^{-13.0}$ | 55.4$^{-30.4}$ | -19.9 |
| SSP (Sorscher et al., 2022) | 87.8$^{-7.8}$ | 82.0$^{-8.7}$ | 60.5$^{-17.8}$ | 73.9$^{-17.4}$ | 41.2$^{-44.2}$ | 15.8$^{-49.9}$ | 74.7$^{-15.9}$ | 55.1$^{-30.7}$ | -24.0 |
| Prune4ReL$^\dagger$ (Park et al., 2023) | 88.4$^{-7.2}$ | 82.2$^{-8.5}$ | 59.7$^{-18.6}$ | 73.3$^{-18.0}$ | 43.5$^{-42.0}$ | 16.2$^{-49.4}$ | 74.2$^{-16.5}$ | 56.1$^{-29.7}$ | -23.7 |
| Dyn-Unc (He et al., 2024) | 92.2$^{-3.4}$ | 86.3$^{-4.3}$ | 79.6$^{+1.2}$ | 90.1$^{-1.2}$ | 86.8$^{+1.3}$ | 47.2$^{-18.5}$ | 81.9$^{-8.8}$ | 71.4$^{-14.4}$ | -6.0 |
| DUAL (Cho et al., 2025) | 93.5$^{-2.1}$ | 90.3$^{-0.3}$ | 80.4$^{+2.1}$ | 91.7$^{+0.4}$ | 86.6$^{+1.1}$ | 45.2$^{-20.5}$ | 90.8$^{+0.1}$ | 75.2$^{-10.6}$ | -3.7 |
| Dynamic Random | 93.0$^{-2.6}$ | 87.4$^{-3.2}$ | 75.9$^{-2.5}$ | 87.4$^{-3.9}$ | 84.6$^{-0.8}$ | 60.2$^{-5.5}$ | 91.4$^{+0.8}$ | 86.8$^{+1.0}$ | -2.1 |
| DivBS (Hong et al., 2024) | 94.8$^{-0.8}$ | 89.8$^{-0.9}$ | 79.8$^{+1.5}$ | 86.2$^{-5.1}$ | 80.3$^{-5.1}$ | 48.5$^{-17.1}$ | 90.0$^{-0.6}$ | 83.8$^{-2.0}$ | -3.8 |
| IES (Yuan et al., 2025) | 94.5$^{-1.1}$ | 89.0$^{-1.6}$ | 76.7$^{-1.6}$ | 86.1$^{-5.3}$ | 74.6$^{-10.9}$ | 35.4$^{-30.3}$ | 87.6$^{-3.0}$ | 73.3$^{-12.5}$ | -8.3 |
| InfoBatch (Qin et al., 2024) | 94.3$^{-1.3}$ | 89.9$^{-0.8}$ | 79.4$^{+1.0}$ | 92.4$^{+1.1}$ | 87.0$^{+1.6}$ | 61.0$^{-4.7}$ | 92.5$^{+1.8}$ | 88.3$^{+2.5}$ | +0.2 |
| **InfoBatch + Ours** | 94.3$^{-1.3}$ | 90.8$^{+0.1}$ | 81.0$^{+2.7}$ | 92.8$^{+1.5}$ | 87.9$^{+2.5}$ | 61.1$^{-4.5}$ | 92.9$^{+2.3}$ | 89.2$^{+3.4}$ | **+0.8** |
| SeTa (Zhou et al., 2025) | 94.3$^{-1.3}$ | 89.5$^{-1.2}$ | 77.7$^{-0.6}$ | 92.2$^{+0.9}$ | 87.1$^{+1.7}$ | 30.4$^{-35.2}$ | 92.2$^{+1.5}$ | 88.1$^{+2.3}$ | -4.0 |
| **SeTa + Ours** | 94.5$^{-1.1}$ | 90.0$^{-0.7}$ | 79.4$^{+1.1}$ | 92.6$^{+1.3}$ | 87.7$^{+2.3}$ | 58.0$^{-7.6}$ | 92.5$^{+1.9}$ | 88.5$^{+2.7}$ | **+0.0** |

Table D: **Classification results of re-labeling integration on CIFAR-100N with ResNet-18.** We combine AlignPrune with SOP+ (Liu et al., 2022a) and compare it against re-labeling-augmented pruning baselines. Performance gaps relative to the full-training setting are indicated as superscripts. The mean $\Delta$ is computed across all noise types.

| Noisy Type → | Real | Symmetric | | | Asymmetric | | Mean |
| --- | --- | --- | --- | --- | --- | --- | --- |
| Pruning Method ↓ | | 0.2 | 0.5 | 0.8 | 0.2 | 0.4 | $\Delta$ |
| Full-training | 56.1 | 71.4 | 58.6 | 39.8 | 72.4 | 63.3 | – |
| **Prune Ratio ~30%** | | | | | | | |
| Prune4ReL (Park et al., 2023) | $64.4^{+8.3}$ | $73.0^{+1.7}$ | $68.2^{+9.6}$ | $22.3^{-17.5}$ | $72.5^{+0.1}$ | $65.9^{+2.6}$ | +0.8 |
| InfoBatch (Qin et al., 2024) | $66.5^{+10.4}$ | $74.0^{+2.6}$ | $68.9^{+10.3}$ | $21.5^{-18.3}$ | $74.6^{+2.2}$ | $71.7^{+8.4}$ | +2.6 |
| **InfoBatch + Ours** | $67.5^{+11.4}$ | $76.3^{+5.0}$ | $70.4^{+11.8}$ | $32.5^{-7.3}$ | $75.8^{+3.5}$ | $72.9^{+9.6}$ | **+5.7** |
| **Prune Ratio ~50%** | | | | | | | |
| Prune4ReL (Park et al., 2023) | $62.2^{+6.1}$ | $69.9^{-1.5}$ | $63.5^{+4.9}$ | $23.8^{-16.0}$ | $68.0^{-4.4}$ | $61.1^{-2.1}$ | -2.2 |
| InfoBatch (Qin et al., 2024) | $66.8^{+10.7}$ | $75.2^{+3.8}$ | $69.0^{+10.4}$ | $24.0^{-15.8}$ | $74.3^{+1.9}$ | $71.6^{+8.4}$ | +3.2 |
| **InfoBatch + Ours** | $67.6^{+11.5}$ | $76.2^{+4.9}$ | $71.6^{+12.9}$ | $32.9^{-7.0}$ | $75.8^{+3.4}$ | $73.1^{+9.8}$ | **+5.9** |
| **Prune Ratio ~70%** | | | | | | | |
| Prune4ReL (Park et al., 2023) | $56.9^{+0.7}$ | $62.9^{-8.5}$ | $57.1^{-1.5}$ | $15.1^{-24.7}$ | $62.5^{-9.8}$ | $54.6^{-8.7}$ | -8.7 |
| InfoBatch (Qin et al., 2024) | $65.8^{+9.7}$ | $74.5^{+3.2}$ | $69.4^{+10.8}$ | $20.5^{-19.3}$ | $74.2^{+1.8}$ | $70.7^{+7.4}$ | +2.3 |
| **InfoBatch + Ours** | $66.0^{+9.9}$ | $76.4^{+5.0}$ | $70.5^{+11.9}$ | $30.7^{-9.1}$ | $75.2^{+2.8}$ | $72.3^{+9.0}$ | **+4.9** |

the best performance across all pruning ratios and noise types, demonstrating strong robustness and generalizability under diverse noisy-label conditions.

**Results with Re-labeling under Label Noise.**   We further explore the integration of AlignPrune with robust learning techniques designed to mitigate label noise. As highlighted in Park et al. (2023), re-labeling strategies can improve model performance by explicitly correcting noisy annotations. We adopt SOP+ (Liu et al., 2022a), a recent method designed to model label noise by disentangling noisy and clean labels through a trainable over-parameterized consistency module and self-regularization, into our framework. Specifically, we use the original hyper-parameter settings from SOP+, and implement it on top of InfoBatch (Qin et al., 2024) and Prune4ReL (Park et al., 2023) to represent state-of-the-art dynamic and static pruning approaches, respectively. As shown in Tables D and E, SOP+ improves all methods, and when combined with AlignPrune, it achieves the state-of-the-art performance across benchmarks. These results confirm that AlignPrune remains compatible with existing re-labeling techniques and can further benefit from them under noisy scenarios.

## B.2 ADDITIONAL EXPERIMENTAL ANALYSIS AND DISCUSSIONS

### B.2.1 ANALYSIS ON EFFICIENCY COMPARISON

As reported in Table 3, AlignPrune with InfoBatch can achieve better classification accuracy while reducing the total training time compared to vanilla InfoBatch. Although AlignPrune introduces a minor computational overhead due to the computation of the Dynamic Alignment Score (DAS) at each epoch, the benefit arises from its improved sample selection efficiency. Unlike InfoBatch, which ranks samples solely based on per-epoch loss, AlignPrune leverages DAS for more robust ranking, enabling more effective identification and pruning of low-utility and noisy samples.

Specifically, as shown in Table F, replacing the original loss-based metric with DAS, and combining with the soft-limit nature of the pruning probability, leads AlignPrune to achieve a higher average pruning ratio per epoch. We argue this is not simply a side effect of more aggressive pruning, but rather a direct consequence of the superior

Table F: **Average ratio of pruned samples per epoch.** Results are reported with ResNet-18 under same prune probability.

| | InfoBatch | **Ours** |
| --- | --- | --- |
| Avg. pruning ratio per epoch | 28.77% | **32.94%** |

sample selection quality of DAS. Because DAS provides a clearer and more confident signal for identifying noisy and uninformative samples compared to raw loss, it can decisively discard a larger

Table E: **Classification results of re-labeling integration on CIFAR-10N with ResNet-18.** We combine AlignPrune with SOP+ (Liu et al., 2022a) and compare it against re-labeling-augmented pruning baselines. Performance gaps relative to the full-training setting are indicated as superscripts. The mean $\Delta$ is computed across all noise types.

| Noisy Type → | Real | | Symmetric | | | Asymmetric | | Mean |
|---|---|---|---|---|---|---|---|---|
| Pruning Method ↓ | Real-A | Real-W | 0.2 | 0.5 | 0.8 | 0.2 | 0.4 | $\Delta$ |
| Full-training | 90.7 | 78.3 | 91.3 | 85.4 | 65.6 | 90.6 | 85.8 | – |
| **Prune Ratio ~30%** | | | | | | | | |
| Prune4ReL (Park et al., 2023) | 94.0 $^{+3.4}$ | 89.7 $^{+11.4}$ | 94.2 $^{+2.9}$ | 91.1 $^{+5.7}$ | 56.5 $^{-9.2}$ | 94.0 $^{+3.4}$ | 92.0 $^{+6.2}$ | +3.4 |
| InfoBatch (Qin et al., 2024) | 94.8 $^{+4.1}$ | 91.1 $^{+12.8}$ | 95.2 $^{+3.9}$ | 94.0 $^{+8.6}$ | 66.8 $^{+1.2}$ | 94.7 $^{+4.1}$ | 93.2 $^{+7.4}$ | +6.0 |
| **InfoBatch + Ours** | 95.1 $^{+4.4}$ | 91.7 $^{+13.4}$ | 95.8 $^{+4.5}$ | 94.6 $^{+9.1}$ | 70.7 $^{+5.1}$ | 95.5 $^{+4.9}$ | 93.5 $^{+7.8}$ | **+7.0** |
| **Prune Ratio ~50%** | | | | | | | | |
| Prune4ReL (Park et al., 2023) | 92.6 $^{+1.9}$ | 88.4 $^{+10.1}$ | 92.8 $^{+1.5}$ | 88.7 $^{+3.3}$ | 49.5 $^{-16.2}$ | 92.9 $^{+2.2}$ | 89.9 $^{+4.1}$ | +1.0 |
| InfoBatch (Qin et al., 2024) | 94.9 $^{+4.2}$ | 90.0 $^{+11.7}$ | 95.0 $^{+3.7}$ | 92.8 $^{+7.4}$ | 66.8 $^{+1.1}$ | 94.7 $^{+4.1}$ | 92.0 $^{+6.2}$ | +5.5 |
| **InfoBatch + Ours** | 95.0 $^{+4.4}$ | 90.0 $^{+11.7}$ | 96.0 $^{+4.7}$ | 94.7 $^{+9.2}$ | 71.1 $^{+5.4}$ | 95.3 $^{+4.6}$ | 93.2 $^{+7.4}$ | **+6.8** |
| **Prune Ratio ~70%** | | | | | | | | |
| Prune4ReL (Park et al., 2023) | 90.2 $^{-0.5}$ | 84.0 $^{+5.7}$ | 90.2 $^{-1.1}$ | 83.3 $^{-2.2}$ | 42.8 $^{-22.8}$ | 89.8 $^{-0.8}$ | 84.0 $^{-1.8}$ | -3.3 |
| InfoBatch (Qin et al., 2024) | 94.0 $^{+3.3}$ | 87.7 $^{+9.4}$ | 94.5 $^{+3.2}$ | 91.3 $^{+5.9}$ | 51.4 $^{-14.3}$ | 92.4 $^{+1.7}$ | 90.4 $^{+4.6}$ | +2.0 |
| **InfoBatch + Ours** | 94.0 $^{+3.3}$ | 87.2 $^{+8.8}$ | 95.1 $^{+3.8}$ | 92.2 $^{+6.8}$ | 58.5 $^{-7.2}$ | 94.8 $^{+4.1}$ | 91.7 $^{+5.9}$ | **+3.7** |

fraction of the dataset at each epoch without harming performance. Thus this improved selection efficiency is what drives both the accuracy gains and the reduction in total training time.

### B.2.2 ANALYSIS ON TRAJECTORY WINDOW SIZE

We provide more results on the effect of trajectory window size $N$ below. We expand the ablation study in Fig. 2b to include smaller (2, 3, 4) and larger window sizes (30, 40, 50) as shown in Table G. Results show that the proposed AlignPrune performs consistently with window size ranging from 4 to 50 across all noisy-label types, with minimal variance presented. However, when using extremely small window size 2 or 3, the performance show significant drop, where the loss trajectory under this condition cannot fully capture the learning dynamics as expected. This confirms that the necessity of such trajectory window design, but also shows that AlignPrune remains robust across a practical and appropriate range of values.

Table G: **Ablation on AlignPrune with varying window sizes.** We evaluate AlignPrune using window sizes ranging from 2 to 50, aiming to validate the necessity of such trajectory window design.

| Noisy Type → | Clean | Real | Symmetric | | | Asymmetric | |
|---|---|---|---|---|---|---|---|
| Window Size $N$ ↓ | | | 0.2 | 0.5 | 0.8 | 0.2 | 0.4 |
| 2 | 77.3 | 55.2 | 70.3 | 58.8 | 40.3 | 71.4 | 63.2 |
| 3 | 78.0 | 55.9 | 70.7 | 59.2 | 40.8 | 71.2 | 63.1 |
| 4 - 50 (avg ± std) | 78.9 ± 0.2 | 56.5 ± 0.3 | 71.3 ± 0.3 | 60.4 ± 0.3 | 41.4 ± 0.3 | 72.2 ± 0.2 | 64.3 ± 0.3 |

### B.2.3 ANALYSIS ON DEPENDENCE OF CLEAN DATA

We provide further discussions on the intuitions behind the effectiveness of AlignPrune with extremely limited or even without clean references.

**Discussion on Reference Set Scale.** From an intuitive perspective, DAS measures the degree of alignment between a sample's loss trajectory and the principal learning dynamics of the target task captured by the clean reference set. Even when the clean reference set is small, its averaged trajectory remains a stable indicator of the model's intended learning trend, because correlation used during DAS computation focuses on *relative trends* rather than absolute loss magnitudes, which is robust to scale differences. To provide evidence from results,

- Results in Fig. 3 show remarkable stability: with only 0.1% clean fraction, DAS provides enough signal to prune noisy samples while preserving informative ones, except under extreme noise (symmetric noise with 0.8 rate), where the clean signal becomes too weak.

- Table 5 provides a comparison with *Clean Validation-set* versus *Noisy Train-set* as the reference dataset. Performance remains consistent for clean-label setting but degrades significantly for noisy labels when using noisy reference data. This confirms that clean reference data is crucial for providing the accurate principal learning dynamics.

In summary, these results indicate that DAS does not depend heavily on the quantity of clean data, but rather on its ability to provide a reliable signal of the principal learning dynamics.

**Discussion on Reference Set Quality.** Using a pseudo-clean set as reference remains effective because a coreset selection method can *filter* the noisy dataset to produce a subset whose average learning dynamic is statistically closer to the true clean dynamic, which can be reflected from the downstream performance of the selected coreset. Even if this reference set is not perfectly purely clean, it has a lower noise level compared to the original noisy set, making its average trajectory a more reliable signal. To provide evidence from results,

- Table 5 shows that using high-quality pseudo-clean subsets (estimated by SmallL or Moderate) yields performance nearly identical to using a real clean reference set, demonstrating the practical viability of this approach in scenarios when clean data is unavailable.

- Table 5 also provides a comparison with *Noisy Train-set* versus *Pseudo Clean-set*. Performance degrades significantly under noisy scenarios, which further confirms that the quality and correctness of the reference set, not just its availability, are critical.

In summary, these results demonstrate that AlignPrune's effectiveness is not contingent on a purely clean reference set. Instead, it highlights the method's practical flexibility: it can remain effective provided with a coreset method capable of extracting a pseudo-clean subset that is reasonably pure to offer a reliable reference trajectory.

### B.2.4 ANALYSIS ON EFFECTIVENESS OF ALIGNPRUNE UNDER CLEAN SCENARIO

By definition, DAS of one sample measures the degree of alignment between its loss trajectory and the average trajectory of the reference set, reflecting whether the sample has the synchronized behavior as the clean reference samples. Additionally, reference samples typically reflect the principal learning dynamics of the target task. Thus, a sample with high DAS exhibits behavior that is more synchronized with principal learning dynamics, suggesting that it is more *informative* and *contributes meaningfully to generalization*.

Under clean-label scenario, this behavior implies that AlignPrune naturally degrades into a surrogate of conventional dynamic pruning method. In other words, when noise is absent, DAS serves as a proxy for identifying informative samples in a way that is consistent with established pruning methods such as InfoBatch or SeTa. This dual behavior, robust under noisy-label and consistent under clean-label, supports the versatility of AlignPrune across both noisy and noise-free settings.

### B.3 ADDITIONAL VALIDATION BEYOND IMAGE

To further demonstrate the broad applicability of AlignPrune, we evaluate it on the NEWS dataset from Kiryo et al. (2017) for text classification. We follow the experimental settings in Yu et al. (2019) to inject the symmetric and pairflip label noise with rates 0.2, 0.5 and 0.45 respectively. As shown in Table H, Align-Prune consistently outperform baseline methods across various types of label noise, which confirms its effectiveness in different modalities beyond image domain.

Table H: **Classification results on NEWS with 3-layer MLP.**

| Noisy Type → | Clean | Symmetric | | Pairflip |
|---|---|---|---|---|
| Pruning Method ↓ | | 0.2 | 0.5 | 0.45 |
| Full-training | 42.5 | 37.1 | 26.7 | 26.5 |
| InfoBatch | 42.8 $^{+0.3}$ | 36.7 $^{-0.4}$ | 26.2 $^{-0.6}$ | 27.4 $^{+0.9}$ |
| **InfoBatch + Ours** | **42.9** $^{+0.4}$ | **37.9** $^{+0.8}$ | **28.4** $^{+1.7}$ | **27.7** $^{+1.2}$ |

Table I: **Statistics signficance analysis on CIFAR-100N with ResNet-18.**

| Noisy Type → | Clean | Real | Symmetric | | | Asymmetric | |
| Pruning Method ↓ | | | 0.2 | 0.5 | 0.8 | 0.2 | 0.4 |
|---|---|---|---|---|---|---|---|
| InfoBatch | 79.0 ± 0.1 | 56.1 ± 0.2 | 71.4 ± 0.2 | 59.7 ± 0.2 | **41.8 ± 0.1** | 71.9 ± 0.2 | 64.2 ± 0.2 |
| **InfoBatch + Ours** | **79.3 ± 0.1** | **59.4 ± 0.2** | **71.8 ± 0.1** | **66.0 ± 0.2** | **41.8 ± 0.1** | **72.6 ± 0.1** | **68.0 ± 0.2** |
| SeTa | 79.0 ± 0.1 | 55.6 ± 0.1 | 70.2 ± 0.1 | 59.0 ± 0.2 | **41.6 ± 0.1** | 71.4 ± 0.2 | 63.2 ± 0.2 |
| **SeTa + Ours** | **79.3 ± 0.1** | **56.3 ± 0.1** | **70.8 ± 0.2** | **60.5 ± 0.2** | **41.6 ± 0.1** | **71.9 ± 0.1** | **64.3 ± 0.2** |

### B.4 ADDITIONAL STATISTICS SIGNFICANCE ANALYSIS

As mentioned in the experimental setup, each experiment is repeated three times, and we report the average accuracy across the runs. Here, we provide a statistics signficance analysis corresponding to the results in Table 1 under prune ratio of 30%. Results in Table I present the mean accuracy along with standard deviations, demonstrating the statistical reliability and stability of AlignPrune. These results confirm the consistency of performance gains observed with our approach.

## C QUALITATIVE VISUALIZATIONS

To better understand the behavior of dynamic data pruning methods under noisy label scenarios and the motivation behind AlignPrune, we present a series of qualitative visualizations from our experiments on CIFAR-100N.

**Static *vs* Dynamic Pruning under Label Noise.** To highlight the strength of dynamic pruning methods under noisy-label settings, we first show the comparison between static and dynamic data pruning methods. Fig. Aa shows that random dynamic pruning, despite discarding samples uniformly at random, consistently outperforms random static pruning across all noise types. This suggests that the temporal diversity of sample exposure, even without informative selection, can lead to greater robustness against label noise. In Fig. Ab, we compare Prune4ReL (the state-of-the-art static pruning method), InfoBatch (the leading dynamic pruning method), and AlignPrune (ours). Across all noisy-label settings, we observe a consistent and clear accuracy gap: AlignPrune > InfoBatch > Prune4ReL. These results validate the inherent advantage of dynamic pruning and further demonstrate that our proposed alignment-based sample selection significantly improves data pruning under noise.

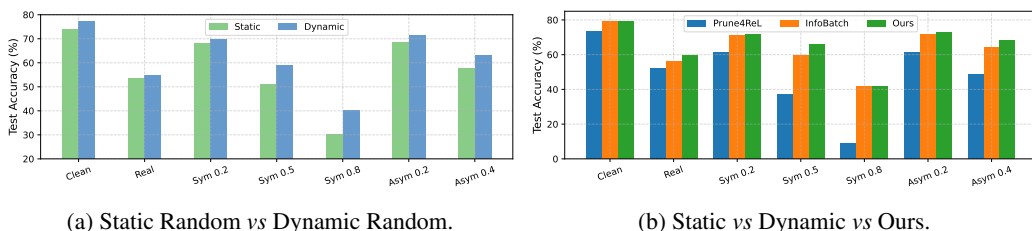

(a) Static Random *vs* Dynamic Random.    (b) Static *vs* Dynamic *vs* Ours.

Figure A: **Performance comparison of pruning methods under label noise.** Best viewed in color.

**Failure of Loss-Based Pruning under Label Noise.** To understand why existing dynamic pruning methods degrade under label noise, we analyze the distribution of training loss for clean and noisy samples. As shown in Fig. B, noisy samples consistently have higher loss values than clean ones. However, methods like InfoBatch discard samples with lower loss values, implicitly assuming these are well-learned and less informative. This leads to a critical issue: the pruning cut-off threshold often lies in a region densely populated by clean samples, resulting in clean samples being pruned while noisy ones are retained. As pruning progresses, the accumulation of noisy examples in the training loop degrades the overall model performance. This observation highlights a key limitation of loss-based selection and motivates our core idea: to replace per-sample loss with a loss-trajectory-based alignment score which can better correlate the clean sample behavior.

**Divergent Loss Trajectories of Clean *vs* Noisy Samples.** To validate the motivation behind our proposed Dynamic Alignment Score (DAS), we visualize the loss trajectories of clean and

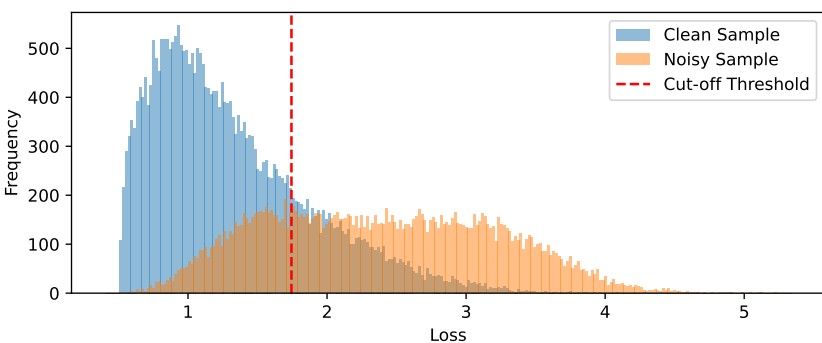

Figure B: **Sample loss distribution of clean *vs* noisy samples.** We visualize the loss distribution of all samples on CIFAR-100N under label noise. Noisy samples tend to exhibit significantly higher loss values than clean ones. Best viewed in color.

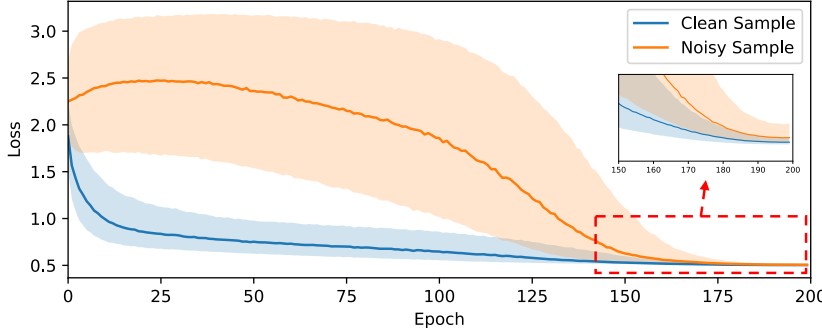

Figure C: **Loss trajectory visualization of clean *vs* noisy samples.** We plot the average loss trajectories over training epochs for representative clean and noisy samples on CIFAR-100N. Best viewed in color.

noisy samples across training epochs. As illustrated in Fig. C, clean samples typically follow a smooth, monotonic decrease in loss, indicating stable learning dynamics. In contrast, noisy samples exhibit non-monotonic loss patterns, often due to label inconsistency or noise over-fitting. This stark difference in trajectory behavior highlights the potential of trajectory-based reference signals in identifying noisy examples. By aligning training sample trajectories with the clean reference trend, DAS provides a reliable and robust criterion for training sample ranking under noisy-label scenarios.

