# OpenReview forum: "AlignPrune: Robust Dynamic Data Pruning through Loss Trajectory Alignment"
_ICLR.cc/2026/Conference — ICLR 2026 Conference Withdrawn Submission_

### Official Review · Reviewer_Wtgn · 2025-10-21

**Soundness:** 1
**Presentation:** 1
**Contribution:** 1
**Rating:** 2
**Confidence:** 5

**Summary:**

The paper targets dynamic data pruning under label noise. The central idea is to replace loss-magnitude–based ranking (vulnerable to retaining noisy examples) with a Dynamic Alignment Score (DAS) that measures the correlation between a sample’s loss trajectory over recent epochs and a clean reference loss trajectory; higher alignment implies a cleaner sample. DAS is plugged into existing dynamic pruning methods without altering architectures or training loops.

**Strengths:**

- Plug-and-play practicality. AlignPrune preserves the base method’s update rule (e.g., InfoBatch’s unbiased gradient rescaling) and requires minimal engineering.

**Weaknesses:**

W1. Limited problem novelty / overlap with prior pruning-under-noise work
The paper positions itself as a study of data pruning under label noise; however, the core problem—robust pruning in the presence of noise—has been actively explored (e.g., Prune4ReL). In this sense, the problem framing is not novel; the incremental novelty is the specific trajectory-alignment criterion and its integration into dynamic pruning. The paper would benefit from clarifying the unique challenges of the setting that Prune4ReL (static) does not cover.

W2. Baseline coverage omits the most directly relevant comparator in the main results
Prune4ReL is listed among related works, but it is not shown in the results. Given that both AlignPrune and Prune4ReL target robust data selection under label noise, readers require a direct comparison of the two.

W3. Presentation quality and missing visual aids
The manuscript’s exposition would benefit from stronger academic polish. In particular: Overlaid clean vs. noisy loss trajectories with their correlation to the reference.

**Questions:**

Please refer to the Weaknesses section.

---

### Official Review · Reviewer_FANS · 2025-10-29

**Soundness:** 3
**Presentation:** 3
**Contribution:** 2
**Rating:** 6
**Confidence:** 3

**Summary:**

This paper studies dynamic data pruning under noisy-label settings.
The author proposes a dynamic alignment score (DAS) that introduces a clean reference subset to evaluate the original dataset.
This method improves the performance of the baselines (InfoBatch and SeTa) in noisy-label settings.

**Strengths:**

1.The data pruning under noisy labels (addressed in this paper) is important and practical.

2.The expression of this paper is clear and logical.

**Weaknesses:**

The innovation of this paper is limited. It seems that only a clean reference set has been introduced based on InfoBatch (or SeTa) to solve the data pruning problem under noisy-label settings. However, there are many important pipeline comparisons missing (such as GLISTER [1], Prune4Rel with Re-labeling [2] on large-scale datasets).

**Questions:**

There are some important questions that the author needs to explain. The specific issues are as follows:

1.The author claims to solve the problem of data pruning under noisy-label settings, but how does this method address the following two important issues？

   (1) How does this method distinguish between hard samples and noisy samples?

   (2) If there are still noise label samples in the selected subset, how does this method prevent overfitting?

2.This paper uses some clean reference samples, but lacks compare with the data pruning method [1] that also uses clean label samples.
[1] GLISTER: Generalization based Data Subset Selection for Efficient and Robust Learning

3.The experimental results in Figure 3 appear suspicious. Why did using as many clean labeled samples as possible as the reference set have little impact on the final results? This is illogical. In noise learning method [3] that uses partially clean labels, the final result will improve as the number of clean samples increases.
[3] Delving into Noisy Label Detection with Clean Data, ICML 2023

4.This paper lacks theoretical analysis. It is hoped that the author analyzes how the number of clean samples affects the final results. From Figure 3, it appears that the impact of the number of clean labels on the performance of the method is not significant.

[1] GLISTER: Generalization based Data Subset Selection for Efficient and Robust Learning. AAAI, 2021.

[2] Robust data pruning under label noise via maximizing re-labeling accuracy. NeurIPS, 2023.

[3] Delving into Noisy Label Detection with Clean Data, ICML 2023.

I hope the author seriously consider and address these issues, as the current score still has room for fluctuation.

---

### Official Review · Reviewer_fkSq · 2025-10-30

**Soundness:** 2
**Presentation:** 3
**Contribution:** 2
**Rating:** 4
**Confidence:** 4

**Summary:**

This paper proposes a dynamic data pruning method for noisy-label settings, which dynamically aligns the original dataset by introducing a clean label set. This method demonstrates consistent performance improvements over baseline methods (InfoBatch and SeTa).

**Strengths:**

1. This paper is well-written.
2. Extensive experiments are conducted on multiple benchmarks.

**Weaknesses:**

1. The novelty is limited. This method introduces a clean $V_ref $in the baselines to solve the noise-label settings. However, this motivation has already been used in reference [1] which is a dynamic method.
2. This method lacks theoretical analysis on how $V_ {ref} $ affects the pruning results.
[1] GLISTER: Generalization based Data Subset Selection for Efficient and Robust Learning. AAAI, 2021.

**Questions:**

1. This method seems to rely heavily on $V_ {ref} $ to distinguish noisy samples, but the results in Figure 3 show that this method appears robust to the size of $V_ {ref} $. I hope the author can provide the proportion of noisy samples in the subset selected by this method under different pruning rates and different sizes of $V_ {ref} $.

2. I doubt about the results in Table B and C (Appendix B). Samples with smaller gradient values are more likely to be clean samples. Since samples with smaller gradient values are more likely to be clean, the SmallL method should theoretically select more clean samples. I hope the author can provide the proportion of noisy labels in the subset selected for each method in Table B.

3. Table D (Appendix B) shows that when using re-labeling, this method outperforms Prune4ReL, which uses subset maximum re-labeling accuracy. Can the author explain why this method is more effective than subset maximization in relabeling accuracy? Since Prune4ReL has theoretical guarantees, its results seem to be reliable and effective.

4. Could the author provide some results to show the number of noise samples in the selected subset?

5. Compare with Reference [1].

---

### Official Review · Reviewer_Aesr · 2025-11-06

**Soundness:** 2
**Presentation:** 3
**Contribution:** 1
**Rating:** 2
**Confidence:** 5

**Summary:**

The paper introduces a dynamic dataset pruning in a noisy label setting. The key idea of the approach is to measure the loss trajectory of the samples and then compare it againt a reference loss trajectory that comes from noise free data. The paper hinges on the assumption that noisy samples exhibit more erratic or inconsistent patterns compared to cleaner samples. Experiments are conducted on standard benchmarks and it is compared against some recent baselines both static and dynamic.

**Strengths:**

Dynamic dataset pruning is very useful idea, particularly for large datasets.
The aproach proposed by the authors is very simple and intuitive.
Experiments are done on standard datasets.

**Weaknesses:**

The major weakness of the paper is that it stands on very simplistic assumptions. While I agree that the loss trajectory of nosiy samples are bit erratic, it can't be conclusivey said, because hard but correct samples also may not smooth loss trajectory. Neural nets hasv the tendency to fit the easy samples first and then the hard examples. Thus, it is very difficult to distinguish if a sample is noisy or hard.

Second comparing against mean loss from a clean validation set is also problematic. Distributional shift could be a real problem in this case. Examples could be nosiy and not noisy, but  non-noisy samples can be also from varying hidden sub-classes. This means a reference score does not reflect who they are. In my view this is going to be a serious issue.

Experimental evaluation is also not up to the mark. Some recent methods like RS2 [Okanovic et al, 2024] or RCAP [Hassan et al, 2025] does very well even with 90% data pruning. Also they are specificaly tested on noisy labels, their performance are very good. Lack of comparison with these most recent works seem major ommission. It may happen that those methods do poorer than the proposed method. But we do not know.

**Questions:**

Please see my weakness section for detailed commnets. These cover my main concerns.

---

### Note · Authors · 2025-11-14

I have read and agree with the venue's withdrawal policy on behalf of myself and my co-authors.